# Heat-Induced Spalling of Concrete: A Review of the Influencing Factors and Their Importance to the Phenomenon

**DOI:** 10.3390/ma15051693

**Published:** 2022-02-24

**Authors:** Hussein Mohammed, Hawreen Ahmed, Rawaz Kurda, Rayed Alyousef, Ahmed Farouk Deifalla

**Affiliations:** 1School of Engineering, University of Edinburgh, Edinburgh EH8 9YL, UK; hussein.mohammed@ed.ac.uk; 2Department of Highway and Bridge Engineering, Technical Engineering College, Erbil Polytechnic University, Erbil 44001, Iraq; hawreen.a@gmail.com; 3Department of Civil Engineering, College of Engineering, Nawroz University, Duhok 42001, Iraq; 4CERIS, Instituto Superior Técnico, Universidade de Lisboa, Av. Rovisco Pais 1, 1049-001 Lisboa, Portugal; 5Department of Civil Engineering, College of Engineering, Prince Sattam Bin Abdulaziz University, Alkharj 16273, Saudi Arabia; r.alyousef@psau.edu.sa; 6Structural Engineering Department, Structural Engineering and Construction Management, Future University in Egypt, New Cairo 11835, Egypt; ahmed.deifalla@fue.edu.eg

**Keywords:** heat-induced spalling, self-compacting concrete, polypropylene fibres, permeability, rate of heating

## Abstract

Heat-induced spalling in concrete is a problem that has been the subject of intense debate. The research community has, despite all the effort invested in this problem, few certain and definitive answers regarding the causes of and the way in which spalling happens. A major reason for this difficulty is the lack of a unified method for testing, which makes comparing data from various studies against each other a difficult task. Many studies have been performed that show the positive effects of using synthetic micro-fibres, such as polypropylene (PP). The mechanism with which PP fibres improve heat-induced spalling resistance in concrete, however, remains a subject of debate. This paper, therefore, looks at the work that has been performed in the field of spalling (particularly spalling of self-compacting concrete (SCC)). Influencing factors are identified and their links to each other (as reported) are discussed. A particular emphasis is put on discussing the role of PP fibres and how they improve the behaviour of high-performance concrete (HPC) at elevated temperatures. A brief summary of the reviewed papers are provided for each of the influencing factors to help the reader navigate with ease through the references. An introduction to heat-induced spalling and the common causes (as reported in the literature) is also included to highlight the wide range of theories trying to explain the spalling phenomenon.

## 1. Introduction

Concrete is the single most widely used construction material in the world [1,2] in all its variety and forms. Generally, concrete is described as a building material that offers a lot in terms of fire resistance [3]. This is a widely used claim when advantages of concrete over other construction materials are advocated.

Despite its benefits, concrete has its disadvantages too; when subjected to an incident heat flux, concrete is shown to exhibit a behaviour known as spalling. Spalling can be defined as the breaking off of concrete layers in either an explosive or a gradual manner when exposed to high temperatures [4]. Spalling is a complicated phenomenon [5] that is still the subject of intense debate amongst the research community [6].

A lot of effort has gone into understanding and mitigating spalling of concrete at elevated temperatures, with a good degree of success. However, there are still some fundamental questions that the research community has not been able to fully answer, even though some of these questions were raised a few decades ago [5,7].

A major obstacle for understanding spalling and mitigating it is the lack of a unified approach for testing [8]. This means that despite an abundance of studies and experimental and numerical results, data from various studies are difficult to compare. Furthermore, with the ever-growing thirst of the construction sector for the use of highly specialised concrete [9,10], the occurrence of spalling has been observed to be remarkably dependant on the type and grade of the concrete that is being put to use [11,12].

In this paper, the authors have tried to summarise the main work that has been performed in the field of spalling, with a particular focus on the factors that are deemed to be highly influential (Figure 1). It is hoped that this will help focus the research community approach to spalling and, ultimately, help understand the problem better.

## 2. Main Hypotheses on Spalling Mechanisms

There are a few hypotheses that describe the causes and mechanisms of spalling. To date, the debate is still ongoing as to which of the mechanisms, mentioned herein, correctly explain spalling and which ones do not [13]. The main hypotheses have been discussed in the next section.

### 2.1. Thermohydraulic Mechanism

Thermohydraulic mechanism refers to the occurrence of spalling due to water vapour pressure. In this theory, vapour forms because of the evaporation of free water and chemically bonded water at elevated temperatures [4]. Due to the gradient of both pressure and molecules that ensues, there will be a movement of moisture and air per Darcy and Fick’s law [14]. The migration of water vapour towards colder inner regions of concrete will lead to the formation of what is referred to as “moisture clog”. This moisture clog acts as a barrier that prevents further migration of vapour into the inner regions of concrete and leads to a pressure build-up [15,16]. The pressure build-up eventually overcomes the (reduced) tensile strength of the heated concrete, which leads to the occurrence of spalling [17,18,19,20,21].

Other researchers [22] have concluded that spalling occurs due to the thermal expansion of liquid water. When heated, water starts evaporating and the pressure starts building up. Since the vapour pressure cannot escape the pores and the pressure continues to rise, this results in having a higher boiling point for the water due to the thermodynamic properties of water (super-heated water). Eventually, cracks start forming in the concrete matrix (due to thermal stresses) and the pressure drops suddenly, leading to instantaneous evaporation of the water and a sudden release of energy that causes spalling.

This is certainly a plausible scenario, but some researchers [23,24,25] observed that the pore pressure was lower in samples that spalled compared to samples that did not; this led them to conclude that pore pressure was not the leading cause for spalling. Other studies show that while pore pressure measurements tend to be higher in samples that are more prone to spalling, the magnitude of the peak pore pressure is considerably less than the reduced tensile strength of the concrete at the point when spalling occurs [26]. This again casts doubt on the validity of the thermohydraulic mechanism as the main cause for spalling.

### 2.2. Thermo-Mechanical Mechanism

Thermo-mechanical mechanism refers to the occurrence of spalling due to mechanical stresses formed because of the temperature gradient when concrete is exposed to a sudden influx of heat. Researchers have recorded the development of a steep gradient of temperature within the concrete subjected to a rapid heating rate [22]. Figure 2 shows the extent of such a temperature gradient when a sample is subjected to a standard ISO 834 fire curve (reported by [22]).

The temperature gradient results in the development of large thermal stresses [27,28]. These stresses would be in the form of compressive stresses at the heated zone (due to the thermal expansion of the concrete) and tensile stresses at the colder regions. Local strain incompatibility between the aggregate and the cement paste also plays a role in this hypothesis; due to the higher temperature, the coarse aggregate dilates while the cement paste starts to shrink due to the evaporation of the water [29]. The combined effects of these occurrences lead to the generation of enough stresses that overcome the tensile strength of the heated sample, at which point spalling occurs.

This mechanism is dependent on the temperature gradient within the concrete sample subjected to thermal loading [27]. As such, a slow heating rate will not be enough to generate the type of temperature gradient required to create enough thermal stresses to cause spalling [22]. Figure 3 (reported by [22]) shows that the in-through temperature is the same as the samples as Figure 1 when a slow heating rate of 1 °C/min is used. The figure shows that the difference between different depths and the heated surface is very small, meaning that a large enough thermal stress due to temperature gradient cannot form.

Researchers have also observed that the addition of small dosages of polypropylene fibres (PP) reduces (or even eliminates) spalling. Due to this observation, some researchers [30,31] have suggested that adding such small amounts of PP fibre is not enough to change the thermal incompatibility problem mentioned earlier. As such, these researchers [30,31] concluded that the thermo-mechanical mechanism alone could not fully explain the spalling phenomena.

### 2.3. Thermo Chemical Mechanism

This mechanism is associated with the chemical degradation of the components of concrete at elevated temperatures. It is generally agreed upon that the strength of concrete decreases when exposed to high temperatures [32,33,34,35,36]. This process starts with hydrothermal reactions where water (free and chemically bound) starts evaporating [4]. At higher temperatures, the chemical bonds between the remaining parts of the concrete matrix start deteriorating [32]. It was observed [37] that the amount of Calcium Silicate Hydrate (C-S-H) within the concrete rapidly decrease as temperatures reach 800 °C. This is mainly due to the breakage of bonds that holds the crystals together. Figure 4 shows the reduction in the value of the characteristic compressive strength for concrete at elevated temperatures [38].

The reduction shown in Figure 3 is widely reported in the literature. However, the exact influence of these changes on spalling are not yet determined.

Researchers [39] found that the presence of moisture adversely affects the mechanical properties of the concrete and, thus, increases spalling propensity. They also found that a larger proportion of C-S-H phases resulted in a more pronounced thermal incompatibility and promoted spalling.

Maier et al. [40] performed a series of tests on concrete slabs (800 mm × 500 mm × 150 mm). The researchers concluded that the main factor for spalling was the increasing pore pressure and not the chemical degradation of the concrete components. The study, nonetheless, stated that deteriorating components of the concrete at elevated temperatures worsened the overall spalling level when it happened.

Whilst conceivable, there are many studies that have observed that the spalling of concrete (especially explosive spalling) happens at an early age of the heating tests when concrete temperatures would be far below the temperatures required for a significant reduction in the concrete strength. Therefore, it is unlikely that degradation of a concrete matrix due to high temperature would be the main reason behind spalling.

### 2.4. Other Hypotheses

Other theories have been proposed by researchers to explain the spalling of concrete at elevated temperatures. These include “the presence and movement of moisture” theory [23]. According to this theory, the presence of moisture at the critical zone (i.e., moisture clog) affects the mechanical properties of the concrete close to the exposed surface. The presence of moisture will also reduce the drying creep, which relaxes thermal stresses. Hence, when a concrete sample has low permeability (to allow for moisture migration and creation of a bigger drying creep zone when heated), the thermal stresses generated will lead to spalling.

Another theory suggests that spalling happens due to the combined effects of both thermal stresses and water vapour pressure [41,42]. Spalling is hypothesised to occur when the sum of all the stresses resulting from rising water vapour pressure and thermal stresses exceed that of the concrete tensile strength.

Researchers have also hypothesised that spalling could be occurring due to fast-moving vapour in concrete when subjected to elevated temperature [43]. According to this theory, water vapour movement creates shear stress along the pore cells. Subsequently, spalling happens when the magnitude of this shear stress exceeds the strength of the concrete. This mechanism, although plausible, has not received much attention and does not have enough evidence to prove its validity [44].

## 3. Types of Spalling

As outlined in the previous section, many researchers are still looking for answers regarding the causes and mechanisms of heat-induced spalling. However, there is a broad agreement that spalling can be categorised into some distinctive types [44,45,46]. A summary of these types are showcased in this section.

### 3.1. Explosive Spalling

Explosive spalling is the most consequential form of spalling. It is referred to as explosive due to its nature; this form of spalling is usually accompanied by a loud bang, which is the result of the explosion-like failure of concrete that is being subjected to an incident heat flux.

Explosive spalling is characterised by its sudden and explosive nature, which is the result of a big and sudden release of energy [46]. Although most spalling does not lead to structural failure [3], recent developments in concrete technology have allowed for the erection of thin, load-bearing elements [12,47]. When explosive spalling happens in such elements, it can lead to its full and sudden collapse [11].

Explosive spalling is observed at an early stage of heat exposure [3,11,43,48,49,50]. Explosive spalling is affected by many factors, such as the properties of the components of the concrete mix, shape, and geometry, presence of load, and environmental factors, such as curing temperature and curing duration [3]. A ranking order of these factors, though, is still not within reach due to the inconclusive nature of the experiments performed so far. Despite this, a lot of researchers have observed that there would be a big reduction in the likelihood of concrete to spall when permeability of the mix is increased. This has led to some researchers concluding that vapour pore pressure was responsible for concrete spalling (relevant research have been outlined in the coming sections). This concept has been disputed by others who have observed that explosive spalling could happen in samples that recorded smaller pressure peaks than samples that did not spall. However, both camps broadly agree that an increase in permeability does lead to improved spalling behaviour of concrete at elevated temperatures.

### 3.2. Surface Spalling

Surface spalling is similar to explosive spalling with the exception that it occurs at the surface of the specimen and is accompanied by a cracking sound [51]. This form of spalling is, depending on the continuity of the heat source, continuous and can affect the performance of a concrete element if it is allowed to continue [43,46]. Ieuan Rickard [8], in his PhD thesis, reported the observation of what he called popcorn spalling. Popcorn spalling was described to happen at very early stages of concrete heating and consisted of many small spallings that lasted for a prolonged period. The author of the study [8] was not able to provide conclusions regarding the causes and mechanism of this form of spalling. However, its properties allow for it to be potentially categorised as a form of surface spalling. Y. Li et al. [31] noticed a similar pattern in their studies where loud spalling bangs were followed by intermittent popcorn-cooking-like sounds, resulting in debris that were less than 5 mm in thickness. The main reason for the concrete to spall like this is yet to be fully determined but considering the temperature gradient between the surface of concrete samples and its inner, cooler regions, this form of spalling may be a direct result of the thermo-mechanical stresses induced by temperature variance at the surface of the sample and the layers immediately underneath the exposed surface.

### 3.3. Corner Spalling

This type of spalling, as its name suggests, happens at the corners of samples exposed to elevated temperatures [3]. Spalling of this type is assumed to occur mainly due to prolonged exposure to elevated temperature, which results in concrete losing its strength [46]. This theory has not been proven, experimentally or otherwise, to be accurate. Like other forms of spalling, corner spalling is very complex, and its causes are perhaps multiple factors culminating in the occurrence of the spalling. Zhao et al. [52] observed that, for cubic samples under fast heating conditions, the thermal stresses induced in the samples started evolving from the corners. These thermal stresses, coupled with the water vapour pressure, can lead to the occurrence of corner spalling, according to the study [52]. The study arrived at these conclusions based on a wide range of assumptions that were made by its authors; therefore it is difficult to say that the conclusions apply to samples without the boundaries of the samples that are considered in the study. Nonetheless, the observation that thermal stresses start from corners makes sense, given that corners have less mass than other parts of the cube and heat up quicker than the rest of the cube [53].

### 3.4. Aggregate Spalling

This form of spalling occurs due to the splitting of the aggregate at the face of the specimen when subjected to elevated temperatures [43]. This is said to be the result of either the mineralogical properties of the aggregate or the thermal shock that aggregate experiences when subjected to elevated temperatures [51]. This type of spalling occurs early into the sample exposed to thermal loading [3].

Aggregate spalling leaves small crates on the surface that were exposed to elevated temperatures. It is therefore considered to not affect the fire resistance properties of the element subjected to thermal load and its effects are only aesthetic [46]. Not many studies have considered this form of spalling or studied why this type of spalling takes place.

## 4. Effects of Type of Concrete on Spalling (Self-Compacting Concrete (SCC)/Vibrated Concrete (VC))

Generally, SCC mixes tend to have a higher propensity to spall than VC [42,54]. This is due to the lower permeability of SCC mixes (compared to VC), which leads to a slower drying rate. Some researchers [55], however, observed that the spalling propensity was influenced by the strength class of the concrete and not the mix type (i.e., SCC or VC).

Noumowe et al. [56] observed that the high strength SCC mixes had a higher propensity to spall, even at a lower heating rate of 0.5 °C/min. The authors suggested that this could be due to utilising calcareous filler in their mix design of the SCC. Having this filler leads to lower permeability of the samples and a higher propensity to spall. It was reported in the study that applying a slow heating rate of 0.5 °C/min resulted in no spalling at all for the samples made using VC, while samples made with self-compacting mix (without fibre) spalled explosively (for each mentioned study, see more details in Table 1).

Anand et al. [57] found that the reduction of strength with increasing temperatures was more severe for SCC concrete than it was for VC when other parameters were kept constant. Similar conclusions were made by Presson [58]. Other researchers [59] observed that SCC mixes were more susceptible to spalling, but that the level of strength loss was more severe in VC than SCC. The authors suggested that if spalling could be controlled (i.e., via the addition of PP fibres), then the residual mechanical properties of SCC mixes would be higher than VC when exposed to elevated temperatures.

Jansson and Boström [54] reported that the probability of high-strength SCC to spall was higher than the corresponding high strength VC. Bakhtiyari et al. [59] also reported that the likelihood of SCC to spall was higher than VC; however, the rate at which mechanical properties of concrete deteriorate at elevated temperatures was found to be steeper in SCC than it was in VC.

The contradictions that exist in the reported literature make it difficult to draw conclusions, but overall, SCC mixes seem to be more prone to explosive spalling due to lower permeability [42].

## 5. Influencing/Mitigating Factors

There are many factors that can, and do, influence the spalling behaviour of concrete subjected to elevated temperatures [5,16,27,60,61,62,63,64]. The influencing parameters can be generally divided into four categories [43]:I.Material: factors related to the properties of the materials used in the mix design of the concrete specimens that are subjected to thermal loading.II.Structural: factors related to the function that the concrete sample is assumed to be serving when in service (i.e., column, beams, slabs, etc.).III.Mechanical: factors associated with the presence of any externally/internally applied loads and the restraint conditions for the sample when it is exposed to elevated temperatures.IV.Temperature: the rate of heating and the max temperature that the sample is exposed to when spalling tests are performed.

These factors are all shown to influence the spalling behaviour of concrete. In the next section, the effects of the most influential factors are discussed as reported in the public literature over the past 15 years, with a particular focus on studies involving self-compacting concrete (SCC). In some locations, older references have been included due to their high relevance to the topic of heat-induced concrete spalling.

### 5.1. Permeability

The permeability of concrete has a major effect on its durability. Researchers have observed that a denser matrix leads to a reduction in the amount of hazardous material that could ingress into the structure of the concrete and cause damages [51].

Permeability is also a big factor in how concrete responds to elevated temperatures; several researchers have noticed that increased permeability almost always leads to a better response as far as spalling is concerned (relevant publications are highlighted in the coming paragraphs). In fact, almost every other factor affecting the spalling of concrete could be viewed from the prism of permeability, as outlined in the next few sections of this paper. Generally, measures that lead to a rise in permeability of concrete at elevated temperatures have a positive impact on the spalling propensity of concrete. One of the most effective ways of mitigating the risks of spalling is through the addition of PP fibres, which most researchers agree improve the spalling behaviour of concrete by increasing its permeability (for each mentioned study, see more details in Table 2). The next section will explain this in more detail.

The permeability of concrete has a major effect on its durability. Researchers have observed that a denser matrix leads to a reduction in the amount of hazardous material that could ingress into the structure of the concrete and cause damages [51].

Permeability is also a big factor in how concrete responds to elevated temperatures; several researchers have noticed that increased permeability almost always leads to a better response as far as spalling is concerned (relevant publications have been highlighted in the coming paragraphs). In fact, almost every other factor affecting the spalling of concrete could be viewed from the prism of permeability, as outlined in the next few sections of this paper. Generally, measures that lead to a rise in permeability of concrete at elevated temperatures have a positive impact on the spalling propensity of concrete. One of the most effective ways of mitigating the risks of spalling is through the addition of PP fibres, which most researchers agree improves the spalling behaviour of concrete by increasing its permeability. The next section will explain this in more detail.

At elevated temperatures, lower permeability may cause explosive spalling due to vapour pore pressure being unable to escape [51] or due to a slow drying front [25] (these concepts are further expanded on later).

Dauti et al. [65] found that the permeability of concrete samples increased with increasing temperature. They found that the rate of the increase in permeability was also affected by the heating rate; the higher the heating rate was, the higher the rate of increase in permeability. The researchers found that axial permeability was influenced by the existence of compressive stresses, even at very low values, such as 0.6 MPa (compared with concrete strength of about 50 MPa).

Peng et al. [66] looked into the effects of curing on spalling. It was observed that combined curing increased permeability of samples and this led to an improvement of their spalling behaviour. The researchers reported that there was little difference in the total amount of pores between samples that spalled and samples that did not. However, it was noted that samples that did not spall had a pore size distribution which was significantly different to samples that spalled. In specimens that were cured using normal curing, pore sizes were either large or small, but in the samples subjected to combined curing (which performed better during spalling tests), pores were refined and there was a better distribution of large, medium, and small pores. This could be the result of having a more connected structure, which enables the movement of moisture at elevated temperatures without causing localised stresses that could lead to spalling.

Kalifa et al. [17] investigated the effects of adding PP fibres to HPC at elevated temperatures. Based on the results, it was observed that the addition of PP fibres improved the performance of concrete at elevated temperatures. The researchers noted that this improvement was accompanied by a big rise in the permeability of the samples that had PP fibre added to them. For mixes with a dosage of 3 kg/m^3^ of PP fibre, it was noticed that the relative permeability (permeability of fibre samples/permeability of non-fibre samples) increased by as much as 85 times at 200 °C. It was also noted that the intrinsic permeability of all samples increased with rising temperature, but the rise in samples with PP fibres was far bigger than samples without fibres. This rise in permeability was accompanied by a big drop in pore pressure for the samples. Maximum pore pressure of 4 MPa was reported for samples without PP fibres whilst pore pressure had a peak value of just over 1 MPa when 3 kg/m^3^ PP fibre was added to the mix. Figure 5 and Figure 6 [17] show the relative increase in the gas permeability of the concrete mixes and the reduction in the value of the peak pore pressure compared to that of concrete mixes without PP fibres, respectively.

Zeiml et al. [67] also looked at how PP fibres improve the spalling behaviour of concrete. The authors concluded that the inclusion of PP fibres results in the rise of permeability of the samples at elevated temperatures. However, their results indicated that this rise in permeability took place at temperatures below the melting point of the fibres. The authors reported that at temperatures below 140 °C, the permeability of concrete samples with 1.5 kg/m^3^ of PP fibres was 3 to 4 times larger than the permeability of samples without PP fibres. According to the authors, this extra permeability, before the melting of fibres in the concrete, is introduced due to the additional interfacial transition zones (ITZ) that are brought about by the fibres.

Miah et al. [21] found that the gas permeability of concrete samples was affected significantly by the applied loads. They reported that the permeability of concrete could either increase or decrease depending on the direction of the compressive load that is applied to the samples. According to the authors, axial permeability increased when preheated samples were loaded axially (parallel to the cracks). Radial permeability, on the other hand, reduced when the axial load was increased for preheated samples up to a certain level of load. After this, permeability stabilised and then increased with a further increase of the load. This is said to be the result of the closing of the cracks (due to the load) and then the opening of the cracks at higher loads due to lateral expansion (Poisson’s effect).

Bosnjak et al. [18] noticed that the addition of PP fibres caused a sudden rise of nearly two orders of magnitude in permeability of concrete samples between temperatures 80 °C and 130 °C. Beyond the range of 130 °C, the rate of increase in permeability was stated by the authors to be almost the same as concrete without fibres. The authors stated that, based on the results, the increase in permeability of fibre reinforced concrete was not related solely to the melting of PP fibres. It was concluded that explosive spalling was prevented in HPC via PP fibres because of the increased permeability.

Ding et al. [68] explored the effect of different fibres on pore pressure for SCC at elevated temperatures. According to the study, pore pressure was reduced significantly in samples with fibres. This is explained by the introduction of extra ITZ around the fibres, which in turn enhances permeability. The increase in permeability was reported to inversely hinder the formation of moisture clog, and this delayed/prevented the occurrence of explosive spalling.

Niknezhad et al. [69] reported that there was no significant difference in the structure between concrete with PP fibre and concrete without PP fibre at room temperature. This changed when the temperature was increased to 500 °C. The SCC with PP fibre showed a greater porous network than concrete samples without PP fibre. This is said to be due to the melting of the fibres that created supplementary porosity in the cementitious paste.

The highlighted studies above all show that measures that result in having a more permeable sample lead to an improvement in the spalling behaviour of concrete at elevated temperatures. The next section shows the impact of adding PP fibres as a way of mitigating spalling.

### 5.2. Addition of Polypropylene Fibre (PP)

It is almost universally agreed upon that the addition of PP fibres is one of the most effective ways to reduce/eliminate spalling in concrete subjected to elevated temperatures. Since the early 1980s, when one of the first reported studies on the effects of the addition of polymers was published [70], numerous studies have been conducted on the effects of the addition of PP fibres into concrete mixes as a way of countering spalling.

The addition of PP fibres to high-performance concrete (HPC) and ultra-high-performance concrete (UHPC) is now included in some design codes in various countries [38]. The Eurocode [38] states that for concrete grades between C80/95 and C90/105, 2 kg/m^3^ PP fibre by weight of cement should be added to the mix to avoid spalling. This value, though, is contested by some researchers who have observed that spalling is still likely even after the addition of 2 kg/m^3^ of PP fibres [71,72]. Furthermore, the code specifies neither the geometry nor the type of the PP fibres to be used, which are shown to significantly affect the spalling behaviour of HPC [48,73,74,75].

Despite its wide usage, the exact mechanism with which PP fibres reduce spalling is not yet fully known [5,6,48]. Some researchers [18,41,51] have concluded that PP fibres release the water vapour pressure (through increased permeability) generated within the concrete when it is subjected to an incident heat flux. However, other researchers [25] have disputed this theory and believe that PP fibres enhance the spalling behaviour of concrete in two ways: first, by reducing moisture in the critical zone that adversely affects the mechanical properties of the concrete, and second, by moving moisture away from the critical zone, which leads to the forming of larger drying creep that releases the thermal stresses within the critical zone.

Below, each of these mechanisms are discussed and the supporting evidence is highlighted.

#### 5.2.1. Pressure Relief Mechanism

The pressure relief mechanism is a theory that has garnered a lot of attention amongst researchers investigating how PP fibres enhance spalling behaviour of heated concrete. According to researchers [18,20,41,51,67,76], PP fibres release the accumulated water vapour pore pressure in concrete in the following ways:

I.Discontinuous reservoirs: because the coefficient of thermal expansion of PP fibres is a few orders of magnitude bigger than that of the surrounding paste, micro-cracks are formed when the sample is heated due to the expansion the PP fibre. These micro-cracks lead to increased permeability, which results in pore pressure being released outside of the sample subjected to elevated temperature.II.Continuous channels: the difference in polarity between PP fibre and water, with PP fibre on one hand and cementitious paste on the other, leads to poor adhesion between PP fibre and the surrounding concrete. Due to this poor interface adhesion, moisture transport can happen via capillary action. PP fibres show thermal instability at temperatures above 80 °C; this is key for this theory to materialise because a weakened PP fibre makes it easier for vapour pressure to escape. Studies [67] have shown that the permeability of concrete samples with PP fibres was 3 to 4 times higher than concrete without PP fibres at temperatures below 140 °C. Since PP fibres melt at about 165–175 °C [77], and the porosity of the samples with and without PP fibres was not significantly different at this temperature range, this further validates the continuous channel theory.III.Vacated channels: at higher temperatures, PP fibres start to melt and then vaporise, leaving behind a series of channels that allow for vapour pressure to escape. This theory is said [41] to be less favourable than the previous ones since the size of these channels is roughly 20–30 µm when the typical capillary pore size within the concrete is 1 µm.

#### 5.2.2. Drying Creep

This mechanism is proposed as an alternative to the pressure release theory. According to this theory [25], PP fibres help to move moisture away from what is defined as the critical zone by the authors. This helps, according to the authors, to improve spalling behaviour of concrete in two ways; (1) the presence of moisture affects the mechanical properties of concrete adversely, and by moving moisture away, the concrete layers at the critical zone retain more of their strength which gives it better spalling resistance; (2) by moving moisture away from the critical zone, the drying creep zone becomes bigger. This helps to release more thermal stresses that were generated by the in-through temperature gradient.

#### 5.2.3. Effects of PP as Reported in the Literature

Researchers have tried to understand the influence of PP fibres on SCC and the mechanism with which PP fibres help to enhance spalling resistance of concrete. Jansson and Boström [25] investigated the influence of pore pressure on spalling. They casted a large number of specimens [78] that ranged from small slabs (600 mm × 500 mm × 200 mm), large slabs, (1800 mm × 1200 mm × 200 mm) and beam samples (3200 mm × 600 mm × 200 mm). Samples had small tubes (2 mm diameter) inserted during casting. Just before the start of the tests, these tubes were filled with silicon oil with a syringe to ensure that no air bubbles were formed at the bottom. The ends of the tubes were connected to a pressure gauge to measure pressure at different depths from the exposed surfaces (for each mentioned study, see more details in Table 3).

The mixes were cast with a W/C ratio of 0.4. Mix SCC B contained a dosage of 1 kg/m^3^ of PP fibres (18 µm diameter) while mix SCC A did not contain any PP fibres. The samples were then subjected to standard (EN 1363-1) fire, modified Hydrocarbon (HCM), and Rijiskwaterstaat fire curves. All the samples were loaded unidirectionally during testing with 10% of their compressive strength.

Results from the tests show that mix SCC B did not spall whereas the SCC A mix experienced spalling. Pressure measurements taken from this test (at different depths from the exposed face) show that the pore pressure was higher in samples that contained PP fibres (i.e., samples that did not spall) except for one case where the reverse was true. These findings can be seen in Figure 7. The authors concluded that, based on their findings, pore pressure was not the main factor behind spalling of concrete.

The researchers published further findings of their report in [60]. They reiterated the positive influence of PP fibres. It was reported that no specimen with PP fibres exhibited any signs of spalling except for one sample, which showed some flaking. The authors reported that this was due to loading. The authors reported that PP fibres accelerated the formation of the drying creep in the critical zone, and this worked to destress the concrete in the critical zone. However, regardless of the point that the researchers raise, PP fibres in this case still work by accelerating the moisture migration from the critical zone through enhanced permeability.

Terrasi et al. [79] performed a large experimental programme to investigate the fire behaviour of prestressed Carbon Fibre Reinforced Polymer (CFRP) High-Performance self-compacting concrete (HPSCC). The slabs had three different thicknesses and three different lengths. Three different mixes were used for the slabs. Mix 042 (no PP fibre), Mix 142 (2 kg/m^3^ PP fibre) and Mix 242 (5 kg/m^3^ PP fibre). The samples were tested in a large-scale fire test corresponding to the ISO 834 fire curve. The slabs were loaded in four-point bending to a level that ensured the decompression of the exposed surface of the slab (i.e., the level of the applied load was determined based on the amount of prestressing within the slabs at the time of testing). In total, 20 CFRP pre-stressed slabs and 2 slabs prestressed with steel tendons were tested.

It was observed that the samples without PP fibres (Mix 042) all spalled extensively in under 30 min. The authors noted that the addition of PP fibres enhanced the spalling resistance of identical slabs. However, only the samples with 5 kg/m^3^ (Mix 242) did not show any spalling at all. Accordingly, requirements in the Eurocode [38] for High-Performance Concrete (HPC) to have 2 kg/m^3^ of PP fibre seem to be insufficient to fully avoid spalling at elevated temperatures. The geometry of the PP fibres used in this study was not reported by the authors. This finding shows that the threshold reported by Kalifa et al. [17] and the amount of PP fibre required by the Eurocode [38] are not sufficient to mitigate spalling completely.

The findings from the above study [79] were corroborated by further research. Maluk et al. [11] tested five slabs under almost similar circumstances. In the experiments, five prestressed CFRP HPSCC slabs were cast. The slabs were 45 and 60 mm in thickness. The prestressing of the slabs took place via stressing the CFRP tendons to an initial 1000 MPa. Two different mixes were used: Mix #A (2 kg/m^3^ PP fibre, 3 mm long) and Mix #B (1.2 kg/m^3^ PP fibre, 6 mm long). The PP fibres were monofilament fibres that had a cross-section of 32 µm.

The slabs were tested in a single, large-scale fire test using the standard ISO 834 fire curve. The samples were subjected to four-point bending during the test. The level of the load was determined such that the exposed faces of the slabs were decompressed (i.e., σc Bottom = 0 MPa).

It was noted that samples from Mix #A (containing 2 kg/m^3^ PP fibre, 3 mm long) tended to spall explosively (3 out of 2 samples) whereas both slabs belonging to Mix #B did not spall. The researchers stated that this was clear evidence that the geometry of fibre influences spalling of concrete as well as its dosage. The method through which PP fibres enhanced the spalling properties of concrete was not investigated by the researchers.

In a detailed experimental programme, Maluk et al. [48] investigated the effects of PP fibre dosage, geometry, number of fibres, type (by supplier), and surface area of fibres on the spalling propensity of HPSCC. In total, 66 medium-sized, unreinforced samples of dimensions 500 × 200 × 45 mm were tested using the Heat-Transfer Rate Inducing System (H-TRIS), a novel technique that utilises radiant heat panels, which were developed at the University of Edinburgh [80]. Samples were subjected to a variable incident heat flux with a maximum value of 100 kW/m^2^. This thermal load was kept for 60 min unless spalling happened before this time. A mechanical loading rig was used to induce compressive stress into the samples during testing. The maximum applied compressive stress during these tests was 12.3 MPa, based on prior research by the authors [6].

Results obtained by the authors indicated that the inclusion of PP fibres with a smaller cross-section were more effective in reducing the propensity of concrete to spall, although the reason for this was not explored by the authors. As for the length of fibres, the authors reported that mixes with shorter fibres (3 mm) exhibited a higher propensity to spall than mixes with longer fibres (6 or 12 mm). Monofilament and multifilament fibres were reported to reduce the propensity of spalling more than fibrillated fibres, although the authors note that this could be related to the difference in surface area for each type.

The results from this study [48], again, suggest that the dosage of PP fibre is not the only influencing factor for spalling propensity. It was observed that some samples, which included a low dosage of 0.68 kg/m^3^ of PP fibres with a smaller cross-section (18 µm diameter), did not spall, while samples containing as much as 2 kg/m^3^ of fibres with a bigger cross-section (37 µm × 200 µm) did. Further, it was noticed that surface area was an important factor in determining the propensity to spall the tested samples. Specimens that included 2.34 kg/m^3^ and 20 mm long fibres did not spall despite having a low number of fibres and a low total length, but they did have a similar surface area compared to other mixes. This, the researchers suggest, shows the importance of the surface area of fibres in improving the spalling behaviour of concrete.

Sultangaliyeva et al. [74] also found that the geometry of PP fibres strongly influences the yield stress of cement pastes. The authors reported that the yield stress of cement pastes increased generally with the increase of the fibre dosage, but the rate of this increase was significantly influenced by the geometry of the fibres used. For the fibres (length = 6 mm, diameter = 18 µm) the increase in the yield stress is 50% more than the increase obtained when using stockier fibres (length = 18 mm, diameter = 34 µm). Further research by the same author [81] suggested that 12 mm long PP fibres were almost 50% less efficient in preventing spalling than 6 mm long PP fibres. This could be a direct result of the deformation that the fibres experience when the concrete is mixed, resulting in shorter fibres providing a higher percolation than longer PP fibres.

Al Qadi et al. [82] investigated the effects of PP fibres on the residual properties of SCC. In this study, four concrete mixes with varying dosages of PP fibres were tested. The mixes were M 0.00 (no PP fibre), M 0.05 (5% PP fibre by volume), M 0.10 (10% PP fibre by volume), and M 0.15 (15% PP fibre by volume). The authors cast 42 cubes and cylinders using the mixes described. Samples were cured for 89 days using a wet curing method. After this period, the samples were tested in an electrical oven for temperatures 200, 400, and 600 °C. The applied heating rate of the oven was 5–10 °C/min. Two testing regimes were followed: heating for two hours and heating for four hours.

The authors observed that the residual strength (both compressive and tensile) of samples containing PP fibres was greater than samples that did not. The authors concluded that 0.05% PP fibre was the optimum dosage since it led to the greatest amount of retention of concrete’s mechanical properties at elevated temperatures.

These findings, however, contradict other researchers’ works. Uysal and Taniyildiz [83] investigated the effects of PP fibres and other mineral additives on the residual compressive strength of SCC after exposure to elevated temperatures. The researchers cast concrete cubes with and without the addition of PP fibres (2 kg/m^3^ dosage) and evaluated the residual compressive strength after subjecting the samples to a heating rate of 1 °C/min in an electric oven. The samples were subjected to 200, 400, 600, and 800 °C. The samples were cooled down at a rate of 0.4 °C/min after 3 h of exposure to the thermal load. The researchers found that the residual compressive strength of samples containing PP fibres was significantly lower than identical samples without PP fibres. Other researchers [55,84] arrived at similar conclusions. Interestingly, the research results performed in [55] suggest that SCC mixes were less prone to spalling compared to Vibrated Concrete (VC) of the same strength class (no explanation is given with regards to this). This is contradictory to what others have reported [54].

Y. Ding et al. [68] investigated the effects of micro PP fibres and macro PP fibres on the spalling behaviour of SCC. The authors prepared 7 different mixes with each mix containing different dosages of either micro-PP fibre or macro PP fibre. Steel fibres (SF) were also used in some of the mixes. The geometry of the microfibres was (length = 9 mm, diameter = 18 µm), and the macro fibre (length = 45 mm, diameter = 74 µm). The microfibre was added to the concrete mixes in doses of 0.5 and 1 kg/m^3^ while the macro fibre was added at a much larger dosage of 4 kg/m^3^. All the mixes were reported to have a compressive strength between 60 and 70 MPa. From these mixes, small beams (150 mm × 150 mm × 550 mm) were cast and wet cured for 28 days. Small tubes were inserted into the mix during the casting of concrete at depths of 10, 20, and 30 mm away from the exposed face of the specimen. These tubes were filled with silicon oil before testing and the end was attached to pressure sensors to measure the pore pressure at these points. Fire tests were then performed to the standard temperature-time curve outlined in ISO 834. Peak pore pressure values at different depths from the exposed surface are shown in Figure 8.

The researchers [68] conclude that the micro PP fibres play a big role in reducing the pore pressure of the samples subjected to elevated temperatures. It was reported that the inclusion of fibres led to a delay in the formation of peak pressure within the concrete, and that the peak pressure in samples containing PP fibres was much smaller than samples that did not.

From the results obtained in [68], the researchers reported that the inclusion of micro PP fibre with SF leads to the best results for the prevention of spalling. It was also reported that measured pore pressure was lower in all samples containing some form of fibres (either PP or SF). This is because of the introduction of new percolation paths by the fibres, which provides a path for the vapour pressure to escape. These paths are likely due to the percolation of the ITZ of the fibres. The authors also reported that the lowest pressure in all the samples was observed closer to the exposed face. This highlights the importance of heat in the build-up of pressure, according to the authors. The researchers observed a steep decline in the mechanical properties of concrete with rising temperatures but noted that the mixes including a cocktail of fibres retained a bigger portion of their strength compared to plain SCC at similar temperatures. The conclusions of the work noted above match well with the results of the investigation into the effects of the cross section of PP fibres on enhancing spalling resistance reported in [48].

Bangi and Horiguchi [75] studied the effects of fibres on pore pressure. They arrived at conclusions that were similar to those obtained in [68]. They casted several cylinders (diameter = 175 mm, length = 100 mm) and tested them using an electric heating plate that could induce a maximum temperature of 600 °C onto the samples. Based on their findings, the authors concluded that the use of longer PP fibres with a smaller cross-section (length = 12 mm, diameter = 16 or 18 µm) performed better than shorter, stockier fibres (length = 6 mm, diameter = 28 and 40 µm) regarding pore pressure reduction. They also noted that PP fibres performed better than polyvinyl alcohol (PVA) fibres. This is said to be due to the lower melting point of PP fibres compared to PVA and its bonding properties. The authors also noted that organic fibres with a melting point close to or lower than the spalling temperature range 190–250 °C performed better, as far as spalling reduction is concerned.

Noumowé et al. [56] investigated a large number of parameters, such as the effect of PP fibre inclusion, heating rate, and type of concrete (SCC or VC). The authors prepared four mixes: High-Strength Concrete (HSC), High-Strength Concrete with fibres (HSCF), High-Strength Self-Compacting Concrete (SCHS), and High-Strength Self-Compacting Concrete with fibre (SCHSCF). The PP fibre dosage was 2 kg/m^3^. Cylinders (diameter = 160 mm, height = 320 mm) and prisms (100 mm × 100 mm × 400 mm) were cast and air-cured for 90 days before testing. Samples were subjected to both a rapid heating ISO 834 fire curve and a slow heating rate of 0.5 °C/min up to 600 °C.

The authors observed that no samples containing PP fibre experienced any spalling at all. This, according to the authors, is evidence of the benefits of including PP fibre in High-Strength Concrete mixes. They also stated that having PP fibres in the HSSC led to slightly lower mechanical properties at ambient temperature compared to their identical counterpart that did not include PP fibre. Having PP fibre also led to lower residual strength of HSCC samples after they were exposed to elevated temperatures. The authors stated that since PP fibres led to a more permeable mix at elevated temperatures, it is therefore common for such mixes to have lower residual strength after exposure to elevated temperatures.

Lura and Terrasi [72] explored a novel approach to reduce/eliminate spalling by adding a cocktail of super absorbent polymers (SAP) and PP fibres together. They hypothesised that the SAP particles (with a diameter ranging between 100 and 300 µm) form water-filled voids. These voids, when the sample is heated, will become void of water, and will form a pore system that allows for the percolation of the ITZ of the PP fibres at lower doses. This is similar to air entrainment agents, but the authors note that the addition of SAP (instead of air entrainment) has the advantage of pore stability and compatibility with superplasticisers and fly ash in the concrete mix.

The researchers cast 6 slab samples (1080 mm × 175 mm × 45 mm) with each slab having 4 CFRP prestressing tendons in the middle of the section. These tendons were stressed to between 1000 and 1200 MPa, creating a central compressive prestress between 7.6 and 105 MPa. Samples were cured for one week in 90% relative humidity (RH) and then for 6–8 months at 50% RH before testing. Fire tests were performed using a small furnace at EMPA laboratories in Switzerland. The time-temperature curve used for this study was the standard ISO 834 fire curve.

The results from the study show that both slabs, which included only PP fibres (2 kg/m^3^ dosage), experienced spalling (at 15 min and 18 min from the start of the test) while slabs containing a mix of both SAP + PP did not show any signs of spalling at all. The authors note that the addition of SAP did reduce the compressive strength of the concrete by about 12% at 90 days. This was explained by the authors to be because of the extra porosity that is generated within the concrete matrix due to the inclusion of SAP. The authors note that this reduction could be overcome by optimising the amount of SAP (1.93 kg/m^3^ were used in this study) and the size of the SAP particles used.

Bosnjak et al. [18] also reported a sudden rise in permeability of samples that included PP fibres when the concrete temperature was ranged from 80 to 160 °C, which is below the melting temperature of PP fibres. The authors explained this as follows: PP fibres have a melting temperature of 171 °C but exhibit thermal instability at about 120 °C. This is accompanied by a reduction in elastic modulus of the fibres, and they become soft. Further, due to the nature of PP fibres, they expand transversely to the direction of the fibre while shortening in length when exposed to heat. A combination of these actions, the authors concluded, leads to the weakening of the bond between the fibre and the surrounding cement paste. This, in turn, leads to a rapid rise in the magnitude of the gas permeability. Due to this, vapour pressure can find a path and move past the not-yet molten PP fibres. These findings further solidify the continuous channel theory that was discussed at the start of the section.

Xargay et al. [85] investigated the residual mechanical properties of high-performance SCC with and without the inclusion of fibres after they had been subjected to elevated temperatures. The authors casted cylinders (diameter = 100 mm, height = 200 mm) and beams (150 × 150 × 600 mm) from 2 different mixes: one was plain SCC and the other had a mixture of steel and PP fibres. Samples were cured at room temperature for 28 days at 90% RH before testing. Testing was performed using an electric furnace. Mechanical properties of samples were tested at 20 °C and after being exposed to 300 °C and 600 °C. The authors reported that the addition of fibres led to samples retaining a bigger portion of their mechanical properties after exposure to higher temperatures compared to samples that did not have fibres in them. Further, the authors noticed that the addition of fibres also led to better integrity of the samples post-exposure to 600 °C and post cracking.

Zhang et al. [86] also investigated the effects of PP fibres on permeability and spalling. They used disc-shaped concrete samples (diameter = 150 mm, height = 45 mm) made using UHPC for gas permeability measurements. They also casted cylinders (diameter = 50 mm, height = 100 mm) to investigate the spalling behaviour of the mixes. The concrete mixes had a compressive strength between 148 MPa and 159 MPa. A 3 kg/m^3^ dosage of PP fibres (length = 12 mm, diameter = 33 µm) was used for the mixes with fibre in them. Discs were heated at a rate of 1 °C/min to 100, 150, 200, and 300 °C. For the spalling tests, the standard time-temperature curve per ISO 834 was used.

The authors reported that permeability increased in all samples with increasing temperature. However, there was a sudden rise of two orders of magnitude in permeability for samples including PP fibre when the concrete temperature was between 105 °C and 150 °C. The authors state that this sudden rise in permeability is because of a network of micro-cracks forming at the fibre-matrix interface due to thermal expansion incompatibilities between the fibres and the surrounding cement paste. Other researchers [21], however, found a similar increase in permeability in concrete samples with no fibres in them. They associated this with the possibility of pores becoming accessible due to the removal of free and physically bound water in these pores.

Y. Li et al. [73] investigated the effect of the geometry of PP fibres on spalling through permeability. The authors found that increasing the length of the PP fibres had a much stronger effect on permeability than increasing the cross-section of the fibre. This was explained by the greater percolation that longer fibres would help to form compared to shorter and thicker fibres.

PP fibres, based on the studies highlighted above, improve the spalling behaviour of concrete at elevated temperatures. Most of this improvement comes from the extra permeability that can be measured when PP fibres are added to the mix of concrete samples at elevated temperatures. No conclusive answer has been found yet regarding how this extra permeability comes about (i.e., is it due to the melting of the fibres, ITZ of the fibres and the surrounding matrix, or the micro-cracks that are formed in the surrounding cement paste because of the thermal expansion of the fibres?). Furthermore, the questions of the dosage of the fibre and the effects of the geometry of the fibre are other parameters that need to be addressed. The dosage of fibres, particularly, is very important given the adverse effects of PP fibres on the flow properties of SCC, as discussed in the next section.

#### 5.2.4. Effect of PP Fibres on Workability

It is generally accepted that the addition of PP fibres to concrete mixes reduces workability. Many researchers [11,17,69,79,82,83,85] have observed this phenomenon. Further, researchers [48] state that the geometry of the fibre used also affects the workability of concrete mixes. According to the results of tests performed on fresh concrete, it was observed that mixes with 0.68 kg/m^3^ of 18 µm diameter PP fibres had a slump flow of 740 mm while mixes that included 1.2 kg/m^3^ of 32 µm diameter PP fibres had a slump flow of 745 mm. This shows that the geometry, as well as the dosage, of fibres has a strong influence on the workability of concrete mixes made using such fibres.

The reason why PP fibres adversely affect the workability of concrete mixes could stem from the fact that flexible fibres (such as PP fibres) tend to bend when they are mixed with concrete [87]. This leads to the fibre occupying any remaining voids within the matrix, which makes the mix denser and less flowable.

### 5.3. Water/Binder Ratio

The amount of water used in the mixing of the components of concrete has a significant effect on the spalling propensity of the concrete produced (for each mentioned study, see more details in Table 4). Many studies have been conducted and there is a broad agreement that a higher W/B ratio leads to a lower propensity of spalling. This is mainly because a higher W/C ratio leads to a more permeable and porous mix, which is generally associated with better spalling behaviour in concrete.

One of the earliest studies on this issue, highlighted in [46], looks into the effects of various parameters on the spalling behaviour of concrete. These parameters include W/C, strength, and moisture content. The study provided a monogram, based on the results, which correlated spalling with moisture content and applied stresses.

Morita et al. [76] performed a series of experimental studies using 30 beam samples (500 mm × 400 mm × 3600 mm) made of high-strength concrete. Samples had varying moisture content (4.2–6.1%), and varying strength (21–118 MPa). The testing took place under the time-temperature curve prescribed in ISO 834. A central load of 44 kN was applied during testing.

The researchers concluded that the W/C ratio had a big effect on the spalling behaviour of concrete. The lower the W/C ratio, the higher the degree of spalling. It was reported that spalling would not occur if the W/C ratio could be kept above 0.5 for two-month-old concrete, and 0.45 for one-year-old concrete. This threshold was defined by the researchers without giving the exact circumstances where this threshold would be applicable.

Hager et al. [88] performed a series of experimental tests using 7 different mixes. The parameters studied were W/C ratio, type of aggregate, and type of cement used. The W/C ratio that was considered in these tests were 0.3, 0.45, and 0.6. Slabs (1200 mm × 1000 mm × 300 mm) were cast, and they were cured in dry air for 90 days. ISO 834 fire curve was used for these experiments.

The authors stated that the W/C ratio had a significant effect on spalling. They found that no spalling occurred in samples with a W/C ratio of 0.6. Spalling was observed to take place in both other samples, but the level of spalling was bigger for samples with a W/C ratio of 0.3 compared to a W/C ratio of 0.45. According to the authors, this is the result of having denser concrete mixes when a higher W/C ratio is used during mixing.

Boström [89] found that there was a linear relationship between the water to powder ratio and the amount of spalling. Results were varying in severity depending on how the samples were cured, though. Further research by the same author [61] showed similar results. However, the author noted that the effect of water/powder ratio did not seem to be as pronounced as reported in the literature.

### 5.4. Type of Aggregate

In an experimental campaign performed by Hager et al. [88], it was found that the amount of spalling was reduced by nearly half when basalt and granite aggregates were used instead of riverbed aggregate.

Khoury [3] suggests that the propensity of thermal explosive spalling is reduced if low thermal expansion aggregate is used. A hierarchy is then provided by the author for aggregates in order of their effect on spalling. Lightweight, basalt, and limestone aggregate is said to be less prone to spalling if used in concrete rather than siliceous and River Thames gravel aggregate. The author emphasises the fact that the hierarchy applies to relatively dry aggregate only. Saturated lightweight aggregate is said to be prone to explosive spalling. Other researchers, however, reported [90] that the type of aggregate had no clear influence on the spalling behaviour of concrete samples. This could be because the authors for [90] considered only two types of aggregate, namely basalt and granite aggregate.

Wu et al. [91] observed that SCC mixes made using lightweight aggregates retained more of their mechanical properties after exposure to elevated temperatures compared to normal weight mixes. However, the use of PP fibres with the mixes makes it difficult to draw any firm conclusion regarding the effects of lightweight aggregates (for each mentioned study, see more details in Table 5).

### 5.5. Size of Aggregate

The results showed that samples containing small- and medium-sized aggregates all spalled at 380 °C while samples containing larger aggregates did not experience any spalling at all. The authors explained the improvement in spalling behaviour of concrete with the increase in aggregate size because larger aggregates have a longer fracture process than smaller ones. This leads to improved spalling behaviour of heated concrete samples, according to the authors (for each mentioned study, see more details in Table 6).

Pan et al. [92] studied the effects of aggregate size on the spalling propensity of concrete. Three different sizes of aggregate were used, namely 2.36–4.75 mm, 4.75–10 mm, and 10–14 mm. Cylinders were cast, and they were cured under different curing regimes. Samples were tested at the ages of 28 and 210 days. Samples were subjected to 800 °C for a duration of 5 h.

Y. Li et al. [31] studied the effects of aggregate size and PP fibres on permeability and pore pressure. They found that utilising PP fibres with larger aggregates increased permeability, mainly due to the thermal mismatch between PP fibres and the aggregate. This mismatch led to the formation of micro-cracks, which increased permeability.

Mohd Ali et al. [90] sought to determine the effects of sample size and aggregate size on concrete at elevated temperatures. Three different aggregate sizes were used; 7, 10 and 14 mm. The results showed that samples with smaller aggregates experienced a greater level of spalling than samples with larger aggregates. Although, the authors noted that the results for the medium-sized aggregates were sporadic and inconsistent.

### 5.6. Concrete Strength/Grade

According to Sideris and Manita [55], spalling likelihood is increased with the increase in the strength of the samples subjected to elevated temperatures. The results of the study show that the tendency to spall is even lower in some of the SCC samples than it is for the VC samples when exposed to the same target temperature (for each mentioned study, see more details in Table 7).

Bakhtiyari et al. [59] conducted tests on samples with compressive strengths between 40 and 50 MPa. They reported that the samples that had higher strength did not spall at 500 °C, which they identified as the critical temperature for spalling. The authors stated that this could be because SF was used for these mixes, which would have increased their residual tensile strength.

Mindeguia et al. [63] experimented with different mixes of concrete to determine the influencing parameters on concrete spalling. The authors found that by increasing the compressive strength of the concrete, the depth of spalling increased. Aslani et al. [93] conducted an experiment using lightweight SCC. During the tests, high-strength and normal strength mixes were used. It was observed that the higher the strength of the concrete, the more prone to spalling it became.

Zheng et al. [49] found that concrete strength was indirectly contributing to an increase in the level of spalling; concrete strength is directly linked to permeability and it is permeability that affects spalling.

Generally, High Strength Concrete seems to be more prone to spalling due to the lower permeability of such mixes [51]. However, it may not be appropriate to categorise spalling using concrete strength since the strength of concrete is dependent on other factors, such as the cement quantity, inclusion of silica fume, and other ingredients.

### 5.7. Externally Induced Stresses

Terrasi et al. [79] observed that spalling happened in the region where there was a minimum bending moment (i.e., near the supports where compressive stress from prestressing would be greatest). This is reported by the authors to showcase the adverse effects of compressive stress on spalling. Such behaviour was also highlighted by other researchers experimenting with simply supported, prestressed concrete elements [94] (for each mentioned study, see more details in Table 8).

Maluk et al. [48] observed that inducing compressive stress resulted in having a higher propensity of spalling for medium-sized, unreinforced samples. No further comment was provided by the author to explain why this was happening. Bosnjak et al. [18] reported that compressive stress led to a slight decrease in the gas permeability of concrete, which led to a rise in the propensity of HPC to spall.

Miah et al. [21] investigated the effects of compressive loading on the permeability of ordinary concrete samples at elevated temperatures. Permeability studies were performed at room temperature on samples subjected to various preloading conditions and preheated to 80, 120, 250, 400, 600, and 800 °C. The authors reported that the presence of a compressive load could lead to an increase or decrease of concrete permeability depending on the direction of the applied load; when the load was in the direction of gas flow (i.e., parallel to the cracks formed at the interface of aggregate and cement paste), permeability increased, but when the load was applied perpendicular to the direction of the gas flow (and the cracks), permeability of the sample decreased. The permeability of concrete also increases in the presence of shrinkage and/or creep due to the occurrence of cracking as, e.g., in PC members, as was reported by Gan et al. [95].

Jansson and Boström [60] found a strong correlation between applied loads and the spalling of concrete slabs. They observed that the effect of the load was more severe in smaller samples than they were in larger ones. The authors noticed that the starting of spalling did not seem to be influenced by the load though; spalling started at similar times for loaded and unloaded samples under identical heating regimes. It was noted that the presence of load, nonetheless, led to a longer period of spalling (in the form of flaking). This was explained by the authors to happen because of compressive stresses closing cracks within the sample that would otherwise help release stresses.

Miah [96] performed a series of tests to determine the effects of load and cement type on spalling of concrete. The conclusion from this study was that the specimens loaded were more prone to spalling than unloaded samples, although there were a few samples that did not experience any spalling even when they were loaded. The author reports that the spalling depth increased with the increase in uniaxial loading. The study found that under low stress, maximum pore pressure tended to increase with the applied stress. For the set-up that was used in this study, it was observed that the depth of spalling was much higher for samples subjected to biaxial loading than samples subjected to uniaxial loading.

Zheng et al. [49] concluded that concrete samples spalled easier in the presence of compressive stress or when the tensile stress was lower on the exposed surface to elevated temperatures. Rickard [8] found that applying a uniaxial load that generated compressive stress equal to 5 MPa led to the occurrence of severe explosive spalling. When the compressive stress was increased beyond this limit, it was noticed that severity of spalling decreased. This observation looks similar to the findings reported in [21] where it was reported that compressive stress beyond a certain point leads to the formation of micro-cracks due to Poisson’s effect.

### 5.8. Heating Rate

Noumawe et al. [56] observed that heating rate influenced the propensity of samples to spall. When a heating rate of 0.5 °C/min was applied, only SCC mixes spalled, and no samples made with VC experienced any spalling. However, when a rapid heating rate (ISO 834) was used, both SCC and VC samples (no fibres included) spalled explosively (for each mentioned study, see more details in Table 9).

Mindeguia et al. [63] noticed that an increase in the severity of thermal loading (i.e., heating rate) caused a decrease in the gas pore pressure in the tested specimens. This was explained by the author to be the result of a damaged concrete surface (as a result of the thermal stresses) which would provide a path for the pressure to escape [97].

Felicetti et al. [98] looked at the influence of pore pressure on spalling using a novel testing method. A concrete cube was placed in the middle of two radiant panels and the pore pressure within the cube was measured using a pressure sensor that was placed within the cube. Four different heating rates were used during these tests (1 °C/min, 2 °C/min, 10 °C/min, and 120 °C/min). According to the results of this study, the heating rate influenced the pore pressure as the following: a quicker heating rate led to faster vaporisation and higher-pressure peaks, but at the same time, thermally induced cracks resulted in higher permeability and pressure release. The authors report that the thermal gradient significantly affected the splitting tensile strength of the samples.

According to Phan [16], higher heating rates lead to the formation of micro-cracks and increased permeability. These cracks allow the pore pressure to escape, thus reducing the maximum measured pore pressure in the sample subjected to such heating regimes. Choe et al. [22] subjected cylindrical specimens to two heating rates: fast and slow heating rates. The authors concluded that fast heating rates resulted in moisture movement due to the thermal gradient because of a fast heating regime. This movement of moisture, according to the authors, led to the formation of a moisture clog that caused spalling. Slow heating, on the other hand, did not lead to the formation of steep temperature gradients. Spalling, in this case, happened due to the boiling liquid pressure within the microstructure of the concrete (BLEVE), according to the study [22].

According to Zhao et al. [52], who numerically analysed cubes (100 mm × 100 mm × 100 mm) subjected to the ISO 834 fire curve and a slow heating rate of 5 °C/min, it was found that the heating rate greatly influenced the way HPC samples spalled. For samples subjected to a rapid heating rate, the temperature gradient induced by rapid heating played a dominant role in the occurrence of explosive spalling, while in samples subjected to a slow heating rate, pore pressure was the dominant factor. It was reported that if the samples subjected to rapid heating could withstand the thermal stresses at early ages, then pore pressure would become the dominant factor in spalling.

### 5.9. Moisture Content/Age of Sample

In Eurocode 2 [38], a limit (k) was prescribed for moisture content. If this limit is smaller than 3%, then spalling is unlikely to occur, according to Eurocode 2. For normal strength concrete, the Eurocode does not mandate any further checks, should this criterion be fulfilled. For concrete grades C55/67 to C80/95, the same rules are applicable if the amount of silica fume is kept below 6% of the weight of the cement. For higher grades of concrete, other measures are prescribed (for each mentioned study, see more details in Table 10).

The above 3% moisture content limit in the Eurocode is somewhat arbitrary. Looking through the literature, there are several instances where concrete samples spalled explosively when the measured moisture content was less than 3% [46,60]. This 3% limit is not likely to be very accurate since the moisture content of concrete varies with thickness. This means that there is more water deep within the concrete layers than there would be near the surface [99].

Jansson and Boström [60] performed a large number of tests on the spalling of SCC. The authors tried to, among other things, assess the influence of ageing and moisture content on the spalling behaviour of SCC mixes. The study showed that the effect of added limestone filler was more dominant than the influence of the W/C ratio for large samples stored for more than two years. The researchers stated that the amount of spalling decreased for samples stored for a longer period. However, spalling was eliminated for only three out of four mixes. Figure 9 and Figure 10 show the maximum and average spalling depths of concrete made with 140 kg/m^3^ and without limestone filler, respectively. It is worth noting that the magnitude of the compressive stress on the samples with no filler was 6.8 MPa while 7.7 MPa was applied for samples with limestone filler in them.

The study [60] showed a strong correlation between moisture content and spalling depth. However, series 39, with the highest amount of limestone filler (140 kg/m^3^), spalled explosively even though the moisture content for this sample was similar to others tested. Figure 11 shows the effect of moisture on the average spalling depths.

Mindeguia et al. [97] looked into the effects of air drying on the spalling behaviour of concrete. Small slabs (700 mm × 600 mm × 150 mm) were cast and subjected to three types of heating rates: Slow heating, medium heating, and rapid heating. It was noted that when concrete samples were pre-dried at 80 °C, no spalling was observed. The study found that pre-drying the samples at 80 °C played a significant role in the reduction of the pore pressure generated when the samples were exposed to elevated temperatures. According to the authors, this showed that free water in the specimens (i.e., moisture content) was responsible for the formation of pore pressure. It was also highlighted that chemically bound water could be released from C-S-H or Portlandlite dehydration since the pressure peak occurred at a temperature where C-S-H dehydration had not started yet.

Maier et al. [40] cast eight concrete slabs (800 mm × 500 mm × 120 mm). Samples were then demoulded and submerged in water for 28 days. After this period, they were put in a heating chamber to obtain a desired levels of moisture content before testing. The study showed that moisture content had a significant effect on spalling; for samples with identical permeability, it was noted that the levels of spalling increased with increasing moisture content. Further, it was noted that for samples with identical moisture content, the level of spalling reduced with increased permeability. The authors concluded that the moisture content and permeability were both significant factors influencing the spalling behaviour of concrete.

Choe et al. [22] cast four different mixes with different W/B ratios (0.55, 0.33, 0.18, 0.12). Samples with a W/B ratio of 0.12 were dried in an oven until their weight stabilised. Then, it was established that the moisture content of the sample was 4.16%. The dried sample, when subjected to elevated temperatures (both slow heating and rapid heating) did not spall at all, despite many visible cracks forming. The undried samples, on the other hand, were destroyed when subjected to elevated temperatures. This, according to the authors [22], highlights the importance of moisture and its movement within the concrete at elevated temperatures.

Peng et al. [66] investigated the effects of combined curing on the mechanical properties of UHPC at elevated temperatures. The results of the study showed that combined curing (using a mix of hot water bath curing and hot dry air curing) led to a significant reduction of the moisture content in the samples from just above 3% in normally cured samples to 1.19% for a combined curing approach. The authors reported that this reduction in free water content, along with improvement of mechanical properties, led to an improvement in spalling behaviour.

Connolly [46] reported tests on loaded normal strength columns (150 mm × 150 mm), which were exposed to elevated temperatures. No spalling was observed when the samples were tested at the age of 2 years when the moisture content was 4.3–4.8%. However, at the age of 5 years, it was reported that spalling started at 12–26 min into the start of the test.

### 5.10. Silica Fume/Binder Ratio

In Eurocode 2 [38], spalling is described as a function of moisture content. The code states that spalling is unlikely to occur should surface moisture content be kept below k (values are given in the national annexes, but the recommended value is 3%). However, for concrete of grades C 55/67–C 80/95, an additional condition is outlined, which is that the maximum content of silica fume should be less than 6% by weight of cement.

In the previous sections, it was explained how the limit on moisture content may not be justified. Nonetheless, this signifies the importance of moisture as far as spalling is concerned. The same could be said for silica fume; whilst it is difficult to tell at exactly what percentage silica fume becomes an issue, its role in the spalling of HPC is important and should be considered carefully (for each mentioned study, see more details in Table 11).

Ahmad et al. [100] investigated the effects of the addition of silica fume to SCC mixes at elevated temperatures and compared the results with identical vibrated concrete samples. The authors reported that the addition of silica fume reduced the workability of SCC mixes when above 2%. This, however, was accompanied by an increase in the yield stress of the mixes. It is also reported that for every 2% addition of silica fume, splitting tensile strength and modulus of rapture was increased by 4% and 5%, respectively. However, the authors reported that samples containing silica fume spalled explosively before reaching 800 °C. This is said to be due to silica fume particles filling the pores of the SCC mix, which led to an increase in pore pressure values.

Bakhtiyari et al. [59] evaluated the effect of having different powders in SCC mixes. Cubes and cylinders were prepared with mixes containing 4% silica fume replacement. Samples were then cured underwater for 7 days and then at room temperature until testing was performed. During the tests, samples were subjected to temperatures of 150, 500, 75, and 1000 °C. The ISO 834 time-temperature curve was used for the study. The authors reported that the inclusion of silica fume resulted in better residual strengths for samples subjected to 500 °C.

Behnood and Ghendehari [35] cast several cylinders (height = 204 mm, diameter = 102 mm) using various mixes. The mixes contained 0%, 6%, and 10% silica fume. Samples were cured in a bath of limewater until the time of the tests. The study found that the compressive strength of the specimens containing 6% and 10% silica fume was increased by 20% and 36%, respectively. The authors explained this by the formation of extra C-S-H, which occupies the pores of the concrete matrix, making it denser and stronger. Upon heating from ambient temperature to 300 °C, at a rate of 3 °C/min, it was noticed that the samples made using 10% silica fume suffered the biggest reduction (31.3%) in strength compared to other mixes. This was reported to be due to the more extensive inner cracking of this mix caused by the vapour pore pressure.

Ju et al. [101] performed an experimental investigation into the effects of silica fume on the spalling of concrete. Five mixes were produced using different amounts of silica fume, which were 0%, 8%, 12%, 16%, and 22%. Cubes (100 mm × 100 mm × 100 mm) were cast using these mixes. Samples were then demoulded and kept at room temperature for 24 h before undergoing steam curing for 72 h. Tests were performed using a slow heating rate of 5 °C/min until a maximum temperature of 500 °C was reached. The study concluded that with the increasing silica fume, the compressive strength increased. This is due to the formation of low permeability C-S-H as a result of the reaction of the CH with SiO_2_ after the initial hydration of the cement. Further, it was noticed in this study that the porosity of the concrete decreased with the increase in silica fume amounts. From the results of this study, it was concluded that adding silica fume to concrete mixes leads to shortening the time required for spalling to start. The authors also reported that the samples with no silica fume spalled into big parts, whereas the samples containing silica fume had their remnants pulverised into a powder. This, according to the authors, is evidence of a dramatic deterioration of stability of the concrete matrix when exposed to elevated temperatures.

### 5.11. Shape of Sample

Al Qadi and Al-Zaidan [82] investigated the effects of sample shape on the residual properties of SCC after exposure to elevated temperatures. The authors cast and compared the results of concrete cubes (100 mm × 100 mm × 100 mm) and concrete cylinders (75 × 150 mm^2^). Samples were cured in water at 20 °C for 89 days before testing. Tests were performed in an electric furnace using a heating rate of 5–10 °C/min and a maximum temperature of 600 °C. the authors reported that cubic samples retained more of their mechanical properties compared to cylindrical samples. This was explained by the authors to be the result of the symmetrical distribution of heat in cylindrical samples compared to cubical samples (for each mentioned study, see more details in Table 12).

According to Du and Zhang [26], the shape of the sample and the number of faces that is exposed to the elevated temperatures influences the amount of pore pressure and subsequently, it influences the spalling behaviour of concrete. For instance, the researchers reported that the probability of spalling for slabs was greater than a beam web subjected to elevated temperatures from three sides.

**Table 12 materials-15-01693-t012:** Heating rate, measurement type, and loading of the scientific research that studied the effects sample shape on the spalling of concrete as reported in the literature.

Paper	Heating Rate	Measurements	Loading
A. Al Qadi, 2014	5–10 °C/min	Visual inspection, residual properties	No
Du, 2020	12 °C/min	Visual inspection, temperature measurements, pore pressure, residual properties	No
Pimienta, 2010	ISO 834	Visual inspection	Yes
Pimienta, 2013	HCM	Visual inspections, spalling depth	No

Pimienta et al. [102] studied the behaviour of large scale walls (2800 mm × 2800 mm × 150 mm), beams (600 mm × 250 mm × 4600 mm) and columns (diameter = 600 mm, height = 2000 mm) made using HPC. The four different mixes were prepared for the study and the compressive strength of each mix was between 89.4 MPa and 111.6 MPa at 28 days. The specimens were subjected to elevated temperature per the ISO 834 fire curve. The researchers reported different patterns of spalling based on the nature of the specimen: for beams, the spalling was very localised, while the entire face of the wall spalled. This was even though all elements belonged to similar mixes and were cured and stored identically. It is worth noting that the elements were loaded during testing; concrete beams were loaded to induce flexure, and columns and walls were loaded uniaxially. The same researchers studied the spalling behaviour of concrete cores embedded in unloaded concrete slabs [103]. The researchers prepared three mixes, namely F1, F2, and F3. Each of the slabs, with the embedded cores, were subjected to ISO 834 fire curve for 2 h. The authors observed that the thermal stresses in the cores were different to their surrounding slabs. The authors concluded that the spalling behaviour of the concrete slabs was different from the spalling behaviour of concrete cores.

### 5.12. Size of Sample

M. Li et al. [104] found that the bigger the size of a sample that is subjected to elevated temperature, the less severe the loss of mechanical properties would be. Jansson and Boström [60] investigated the effects of the size of samples on the spalling of self-compacting concrete. They found that smaller slabs spalled considerably less than larger slabs under identical loading conditions and slab thickness. They explained this to be the result of (1) boundary conditions being a more dominant factor in smaller slabs than they are in larger slabs, (2) release of stresses at the boundaries being more pronounced for smaller slabs than it is for larger slabs, and (3) the presence of reinforcement in the larger slabs that prevented the formation of cracks, which result in higher stress. The authors also reported that smaller samples were influenced more by the presence of compressive stress than larger samples. The authors highlighted previous studies that agree with what was found in these experiments: when a 200 × 200 mm area of a sample (600 mm × 600 mm× 200 mm) was heated from one side, the cooler concrete around the central region acted as a restraint that acted against thermal expansion. This led to spalling until the cooler concrete cracked, at which point spalling ceased immediately (For each mentioned study, see more details in Table 13).

### 5.13. Curing

Curing is another factor that is shown to influence the properties of concrete and, by extension, its spalling behaviour. Peng et al. [66] looked into the effects of combined curing of HPC properties and on its spalling behaviour. Cubes (100 mm × 100 mm × 100 mm) were cast for these tests. Three different curing regimes were followed: (1) wet curing in a water bath at 20 °C, (2) hot water curing at 90 °C, and (3) dry air heating at 200–250 °C for 1–3 days. Some of the samples were subjected to more than one treatment method. Cubes were placed in an electric furnace 28 days after casting and the temperature was raised at 10 °C/min to a maximum of 800 °C. Specimens subjected to combined hot water and dry air curing showed an improved compressive and splitting tensile strength (a 49% and 59% increase compared to samples cured in water at 20 °C, respectively). Furthermore, the authors reported that combined curing led to an increase in the fracture energy of the samples by 10% and 45%, depending on how long the dry air treatment lasted (for each mentioned study, see more details in Table 14).

The researchers [66] found that combined curing had a favourable effect on explosive spalling; samples that were subjected to combined hot water and hot dry air curing experienced significantly less spalling than other samples. When plain samples (i.e., no fibre) that had been cured using a combined method were subjected to elevated temperatures, no spalling was recorded. Samples subjected to hot dry air (250 °C) for 3 days also avoided spalling. The researchers attributed this to the reduction in internal free water due to the higher level of hydration and pozzolanic reaction. This, according to the authors, led to a reduction in the amount of recorded pore pressure. The authors stated that the moisture content was significantly reduced by utilising combined curing: samples cured in a water bath showed moisture content just above 3% compared to 1.19% when samples were cured in hot water for two days and then exposed to hot dry air (250 °C) for three days. This reduction in free water was accompanied by a sharp rise in the hydration degree of cement within the samples. For specimens that were cured in water baths, a hydration degree just above 50% was achieved. However, for samples subjected to combined curing, a maximum hydration degree of 73.4% was reported.

Curing time seems to be of less importance than W/B or the compressive strength of the concrete subjected to elevated temperatures, according to studies performed by Jansson and Boström [60]. The researchers found that some samples experienced severe spalling after they had been stored for four years. This sample was cast from a mix that contained the highest amount of limestone filler (140 kg/m^3^), which the authors suggested might be the reason for the severe spalling. Turkmen and Kantarci [105] experimented with the effects of different curing regimes on SCC properties. They used five different curing regimes (CC1 in lime saturated water, CC2 dry in the air, CC3 wetted 3 times a day for 14 days and then dried in the air, CC4 under a wet sack for 14 days and then cured in air, and CC5 100% constant RH). The researchers found that for samples cured in a bath of lime water, the capillary coefficient (in cm^2^/min) was nearly ten times lower than samples that were cured in dry air. The researchers note that this could be due to higher pore sizes of specimens cured in air conditions, compared to samples cured in water baths. Similar findings were reported by Oliveira et al. [106]. The authors of this study compared the porosity of uncured samples with samples that had been cured for an initial seven days after demoulding. It was reported that samples with seven days of curing in a water bath had a lower porosity than samples that were not cured. Other researchers [107] have observed that maximum compressive strength in SCC could be obtained when samples are cured in water for 28 days, as opposed to a shorter period of curing or no curing at all. Curing increases the elastic modulus and compressive strength of concrete as reported by Singh et al. [108]. Consequently, porosity of concrete is influenced with time.

Generally, there is a trend pointing towards samples having higher strength when cured in water. This could lead to samples developing a denser matrix that is more prone to explosive spalling.

### 5.14. Other Types of Fibres/Additives

Abdulhalem et al. [109] investigated the effects of SF on the behaviour of SCC corbels after exposure to elevated temperatures. The authors concluded that, whilst steel fibres on their own did not prevent spalling, they enhanced the ductility of the concrete samples. It was also reported that the inclusion of steel fibres led to better residual properties for the concrete subjected to elevated temperatures. Nonetheless, the authors reported that the inclusion of PP fibres was still necessary to mitigate the risk of explosive spalling. It was also observed that with rising temperatures, the mechanical properties of the samples dropped significantly. Similar findings were reported by other researchers [110,111] (for each mentioned study, see more details in Table 15).

Ding et al. [112] studied the effects of the inclusion of SF and PP fibre on the residual properties of HPSCC at elevated temperatures. The researchers concluded that the addition of SF alone cannot prevent spalling in HPSCC. The authors reported that higher residual properties were measured in samples reinforced with SF after exposure to heating. It was concluded that higher flexural toughness was observed in samples with a mix of PP fibres and SF when exposed to elevated temperatures. The researchers noticed that a maximum hybrid amount of 55 kg/m^3^ SF and 2 kg/m^3^ PP fibre, or 40 kg/m^3^ SF and 3 kg/m^3^ PP fibre, could be the upper limit of fibre content before self-compacting properties of concrete are lost. Further studies by Ding et al. [68] reported that the maximum reduction in pore pressure for samples made of SCC occurred when a cocktail of PP fibre and SF were used together. The inclusion of SF alone had very minor effects on the reduction of pore pressure in samples subjected to elevated temperatures.

Based on the results from this study [68], the researchers concluded that a cocktail of SF and PP fibre may present the best results for mitigating spalling in concrete at elevated temperatures.

Ozawa et al. [113] looked at the effect of various fibres on the spalling of UHPC. They cast 24 cylinders (50 mm × 100 mm), using mixes that contained varying amounts of PP fibres, Jute fibres, and SF. The samples were steam cured for 48 h at 90 °C. Samples were then subjected to an ISO 834 fire curve for 30 min. The authors noted that the temperature inside the oven was consistently below the ISO 834 curve.

The results showed that PP fibres (length = 12 mm, diameter = 42 µm) were indeed very effective in reducing the amount of spalling. However, the authors reported that the best results were achieved when jute fibres (length = 12 mm, diameter = 10–30 µm) were used at a dosage of 0.5% (by volume). The authors correlated this with the higher melting temperature of PP fibres compared to jute fibres, which carbonise when heated. These results are very limited compared to the vast amount of literature that shows the superiority of PP fibres in reducing spalling. Therefore, further verification of these results would be necessary.

Han et al. [114] investigated the effects of type and contents of polymer resin on spalling of HPC at elevated temperatures. It was reported that samples containing additives experienced spalling after exposure to ISO 834 fire. However, the extent of the spalling was less severe compared to that recorded for the plain concrete samples.

### 5.15. Air Entrainment

Several researchers have studied the effects of air entrainment on the properties of concrete at ambient and elevated temperatures [115,116,117]. As shown in the previous sections, the permeability of concrete samples greatly affects the performance of concrete spalling. The addition of an air-entraining agent reduces the moisture content, and thus reduces the risk of spalling [3]. Air entrainment, however, adversely affects the mechanical properties of concrete, according to Drzymala et al. [116]. In the study [116], the effects of air-entraining agents and the inclusion of PP fibre on mechanical properties of HPC was assessed. It was observed that the mechanical properties of PP fibre-reinforced concrete dropped as well. The authors, however, concluded that air-entraining increased the resistance of HPC to spall at elevated temperatures effectively and feasibly. Therefore, the authors stated, it could be used as an alternative to PP fibre to mitigate the risks of spalling for HPC (for each mentioned study, see more details in Table 16).

Khaliq and Waheed [115] looked at the propensity of spalling of air-entrained HPC. Four mixes were prepared: two with air entrainment (4% and 8%) and two without. Cylinders were cast and the specimens were cured in a water tank for 28 days. Samples were then subjected to elevated temperatures of 100, 200, 400, 600, and 800 °C. An electric furnace with a heating rate of 10 °C/min was used. The authors reported that both non-air-entrained samples spalled at temperatures of 400 °C and 600 °C. Air-entrained samples, on the other hand, did not experience any spalling at all, even when the temperature of the oven was 800 °C.

Lura and Terrasi [72] used SAP along with PP fibres to reduce spalling of HPSCC. The authors noted that the effects of SAP were similar to air entrainment, except that the stability and uniformity of voids could be controlled better with the addition of SAP rather than air entrainment.

## 6. Conclusions

Spalling of concrete is a phenomenon that the scientific community has been trying to solve for the past few decades. The sections above explore the mechanisms, the influencing factors, and the measurement methods that have been used by the research community to investigate spalling and its mitigation.

The main conclusions that could be drawn from this review are as follows:

### 6.1. General

The lack of standard spalling tests has made comparing data obtained by various researchers a difficult task. Whilst there is broad agreement on some of the aspects of the issues, such as the factors affecting spalling, there are disagreements amongst the research community on the ranking order of the influences of these parameters.A few mechanisms for spalling have been put forward, the main ones being the Thermohydraulic Mechanism (i.e., spalling due to vapour pore pressure) and the Thermo-Mechanical Mechanism (i.e., thermal stresses within the heated samples causing spalling). The research community has not been able to fully determine which mechanism can explain the spalling phenomena, with some researchers arguing that spalling is perhaps the result of both mechanisms acting together.Spalling can be categorised into a few types, such as explosive spalling, surface spalling, corner spalling, and aggregate spalling. Explosive spalling is the most destructive type, and its effects can be detrimental to the structure.

### 6.2. Permeability

Permeability and its effects on both spalling propensity and the level of spalling have been studied extensively. Several researchers have reported a direct link between permeability and spalling.Studies have shown that permeability generally increases with an increase in temperature. However, permeability is increased by several orders of magnitude when PP fibres are included.Researchers have also studied the effects of curing on permeability. It has been reported that certain curing conditions led to a refinement of the pore size distribution which led to better spalling resistance.There are different hypotheses on how the addition of PP fibres increases permeability; some researchers report that the melting of PP fibres is what causes the rise in permeability, whilst other studies have concluded that there is a significant rise in permeability of samples with PP fibres before the melting temperature of PP fibres is reached. This is said to be a result of the softening of the fibres, which leads to a deterioration of the interface between the fibre and the concrete matrix, allowing the vapour pore pressure to escape.It has been reported that permeability is affected significantly by applied compressive loads and the direction of the load relative to the measured permeability. In general, applying a load that could result in the closing of micro-cracks leads to a reduction in the gas permeability of concrete.The addition of PP fibres seems to increase permeability due to the additional ITZ that is introduced at the interface between the fibres and the concrete matrix, which enhances permeability.

### 6.3. Inclusion of PP Fibres

PP fibres seem to be an efficient way of mitigating the risk of spalling. The mechanism through which PP fibres improve the spalling behaviour of concrete is still not clear though. It is observed by many researchers that the inclusion of PP fibres increases permeability.Some researchers have concluded that the increase in permeability with the inclusion of PP fibres is due to the melting of PP fibres. This increases permeability to gases and allows pore pressure, generated by the heat, to escape without causing spalling.Other researchers have observed that the increase in permeability occurs before the temperature reaches the melting point of PP fibres. This is due to the decrease in elastic modulus of the fibres, which makes them soft, and gases can escape through the deteriorated interface between the fibres and the concrete matrix.The increase in permeability has been explained by some researchers to come from the micro-cracks forming within the cement paste that surrounds the fibres; due to thermal expansion of the fibres and the thermal incompatibility of the fibres with their surroundings, the cement paste around the fibres will develop a network of micro-cracks that will lead to increased permeability.Studies show that the geometry of the PP fibres has a big influence on both spalling behaviour and the workability of SCC. PP fibres with cross-sections bigger than 50 µm do not seem to have a noticeable impact on the spalling behaviour. In general, thinner PP fibres seem to improve spalling behaviour better, but they have a worse effect on workability.Very short PP fibres (3 mm) perform less well than longer ones, although the longer fibres have a more adverse effect on workability than the shorter ones.The dosage of PP fibres has a big impact on spalling behaviour, with bigger dosages leading to the elimination/reduction of spalling. However, this impact is significantly affected by the geometry of the fibres used, as discussed in points 6–8. The inclusion of bigger dosages of PP fibres also reduces workability significantly.Researchers have also reported inconclusive results for the effect of PP fibres on residual mechanical properties of concrete after exposure to heat. Whilst some results seem to indicate that residual properties were enhanced with the inclusion of PP fibres, other studies show the reverse to be true.Super Absorbent Polymer (SAP) positively affects the spalling behaviour of concrete when used in tandem with PP fibres. It is assumed that the SAP voids, left behind after the evaporation of its water content at elevated temperatures, act as a medium between the links created by the PP fibres and helping to create a continuous network through which the pore pressure could escape before causing spalling.

### 6.4. W/B Ratio

W/B ratio has been reported in some of the literature as having a significant effect on the spalling propensity of samples, mainly due to the compact nature of mixes with a lower W/B ratio. However, other studies point out that, whilst the W/B ratio does have an impact on the spalling propensity, the effects are somewhat less severe than what is reported in the literature.

### 6.5. Type and Size of Aggregate

The effects of type of aggregate on spalling are not entirely clear. Some studies show that low thermal aggregates reduce the propensity of spalling, while other studies show that the type of aggregate had no clear influence on the spalling propensity of concrete samples.The size of aggregate used in the mix of concrete is reported to have an impact on the spalling of concrete samples. Studies have shown that bigger aggregate results in having a more permeable concrete and, thus, reduce the spalling of concrete. Although some researchers have reported that the effect of aggregate size seems to be sporadic and not entirely clear cut.

### 6.6. Type of Concrete/Mix

SCC concrete has been shown by many studies to be more prone to spalling than VC. The reason behind this is due to the lower permeability of SCC compared to VC since the filler in the SCC mix makes the concrete more compactly. Studies have shown that SCC is more prone to spalling, even with low heating rates.However, some studies have reported that spalling is mainly affected by the strength grade of the concrete and not its type.Whilst drawing conclusions is difficult in such circumstances, there seems to be a trend in SCC mixes having a higher propensity to spall.

### 6.7. Concrete Strength

Generally, it has been reported that increasing the strength of samples leads to an increase in propensity to spall.Some studies have shown that concrete strength, not concrete mix type (i.e., SCC or VC), is the important factor in terms of spalling likelihood.It has been reported that by increasing concrete strength, spalling depths also increased.Some studies have suggested that strength is indirectly related to the increase in the level of spalling; increasing concrete strength leads to reduced permeability and this leads to a higher level of spalling.

### 6.8. Externally Induced Stresses

The effects of externally induced stresses (i.e., mechanical loading) on spalling is not fully clear. While, generally, loading seems to increase the propensity to spall, the degree to which this increase is affected by loading is not fully clear. Some researchers have observed that zones subjected to compressive stresses exhibit more spalling whilst areas subjected to tensile stresses show less spalling. It has also been reported that the increase in compressive stresses increase spalling propensity but only up to a certain point, beyond which spalling does not seem to be affected by increasing the load.It has also been reported that the direction of applied loads affects permeability, and by extension, spalling. When pre-heated samples were subjected to axial loads, radial permeability was reported to decrease, but when the applied load was parallel to the radial direction, axial permeability decreased.

### 6.9. Heating Rate

The information available on the effects of heating rate is not conclusive. However, there seems to be a general trend that rapid heating adversely affects the spalling behaviour of concrete at elevated temperatures.It has been reported that rapid heating leads to a reduction in pore pressure due to cracks forming at the exposed surface of the concrete (due to thermal stresses), which allows vapour pressure to escape.According to some studies, Faster heating rates lead to faster migration of moisture towards the centre of specimens, leading to the formation of moisture clogs, which lead to spalling. Slower heating rates, on the other hand, do not lead to big temperature gradients within the samples. Instead, spalling happens due to the rise in boiling temperature of the moisture that is trapped within (BLEVE).The mechanism of spalling is dependent on the heating rate, according to some researchers. Samples subjected to rapid heating rates spall mainly due to the thermal stresses induced by the temperature gradient. Samples subjected to a slower heating rate tend to spall mainly due to vapour pore pressure due to lack of micro-cracks forming because of thermal stresses.

### 6.10. Moisture Content/Age of Concrete

In the Eurocodes, it is advised that spalling would be an unlikely event if moisture content was kept below 3%. Studies performed to try and investigate the effects of moisture content show that this limit is somewhat arbitrary. The code has not specified where within the sample this limit applies (i.e., the surface of the sample or the centre of the sample).Spalling tests on SCC have shown that the effects of filler amount and W/C ratio was more pronounced than moisture content. Samples that had aged for over two years spalled explosively with an increased amount of limestone filler.However, other studies show that moisture content plays a significant role in the spalling reduction of concrete. When samples were dried at 80 °C, no spalling was observed.For samples with identical permeability, it was noted that increased moisture content led to more spalling.Combined curing has been used by some authors to try and reduce the free water within concrete samples. The results of such experiments indicate that by reducing the moisture content in the samples from circa 3% to nearly 1.2%, the samples exhibited much better resistance to spalling when subjected to elevated temperatures.

### 6.11. Silica Fume/Biner Ratio

Eurocode 2 sets the upper limit for silica fume in High Strength Concrete (grade C55/67 to C80/95) as 6% by weight of cement. Results of studies into the effects of silica fume show that the addition of amounts above 2% reduces the workability of mixes. The compressive and tensile strength capacity of the samples was increased, on the other hand.This increase in strength is generally accompanied by an increase in the density and compactness of the concrete matrix. This leads to lower permeability which, in turn, leads to greater levels of spalling.

### 6.12. Shape/Size of Samples

The shape and sizes of samples subjected to elevated temperatures are reported to influence the level of spalling. Tests performed indicate that cubic samples retained more of their strength after being exposed to elevated temperatures compared to cylindrical samples due to the difference in heat distribution.It has also been reported that the number of faces exposed to elevated temperature influences the pore pressure and spalling behaviour of samples. Accordingly, a slab will be more likely to spall than the web of a beam that is subjected to elevated temperatures from three sides.Spalling is reported to be different based on the function a structural member serves; a beam that is made using a similar mix to a wall, was reported to have only localised spalling while the entire face of the wall spalled.Researchers have seen that the bigger the size of a sample, the less severe the reduction tends to be in the mechanical properties after exposure to elevated temperatures.Other researchers have observed that smaller slabs spalled considerably less than bigger slabs under identical conditions. This is said to be because of boundary conditions and the stress release at the boundary conditions which tend to be more pronounced for smaller samples. It was also reported that smaller slabs were more affected by compressive loads than larger slabs.According to the available results in the literature, a firm conclusion could not be made about the effects of size and shape on spalling. However, boundary conditions are influenced by the shape and size of samples, which then affect spalling.

### 6.13. Curing Effects

Dry heating and hot water curing have been observed by some researchers to lead to an increase in both compressive and splitting tensile strength.Combined curing has also led to better hydration of the cement and further reduction on the internal free water. This, considering the influence of moisture content on spalling, is thought to lead to a reduction in spalling.However, some researchers have reported that curing time and curing method has less impact than other factors, such as the W/B ratio. Research has also shown that samples that are cured longer tend to develop a denser matrix, which could make them more prone to spalling.

### 6.14. Other Types of Fibres

Steel fibres, when used together with PP fibres, positively affect the behaviour of concrete via increasing its ductility. It is also noted that steel fibres increase the residual mechanical properties of samples that have been exposed to elevated temperatures. However, available studies show that steel fibres on their own are not enough to prevent spalling.Some researchers noted that the inclusion of steel fibres on their own did not lead to a noticeable reduction in pore pressure. However, when used with PP fibres, steel fibres led to a major reduction in measured pore pressure for samples exposed to elevated temperatures.Jute fibres have shown, by at least one set of experiments, to enhance the spalling resistance of concrete cylinders more than PP fibres. The authors of the study explain this by the lower melting point of Jute fibres compared with PP fibres.

### 6.15. Air Entrainment

Air entrainment generally enhances the spalling resistance of concrete at elevated temperatures. However, air entrainment influences the mechanical properties of concrete adversely. Researchers have noted that the pore stability and uniformity could not be controlled when air entrainment agents are used to mitigate spalling.

## Figures and Tables

**Figure 1 materials-15-01693-f001:**
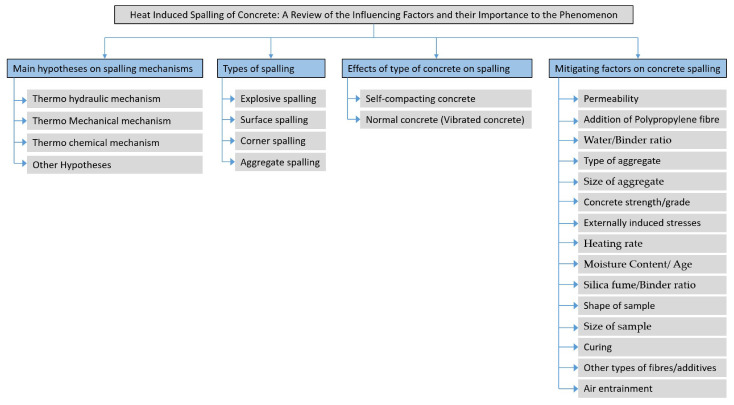
Structure of the paper and studied parameters.

**Figure 2 materials-15-01693-f002:**
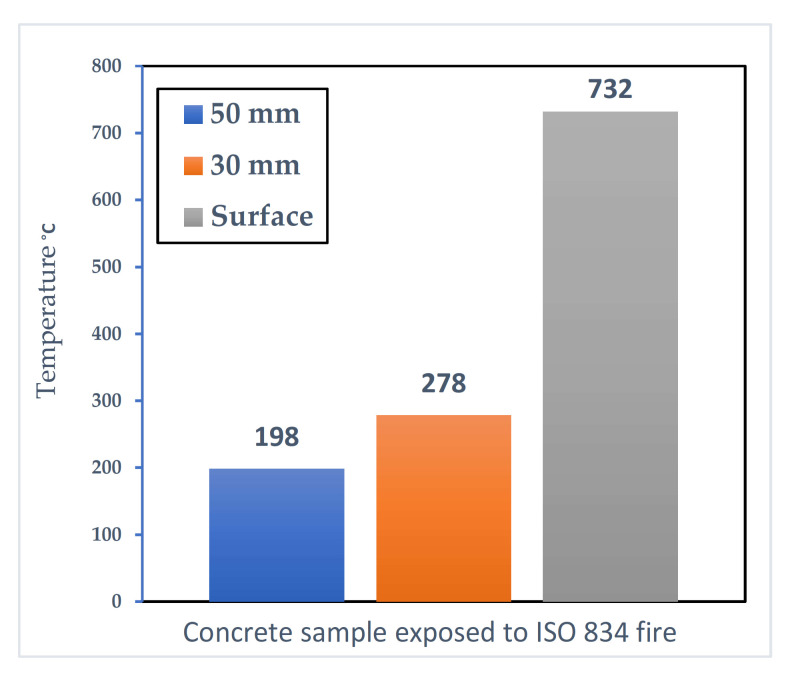
The temperature gradient after 20 min for a concrete sample exposed to ISO 834 fire. Temperatures are recorded at the exposed surface, and 30 and 50 mm from the exposed surface.

**Figure 3 materials-15-01693-f003:**
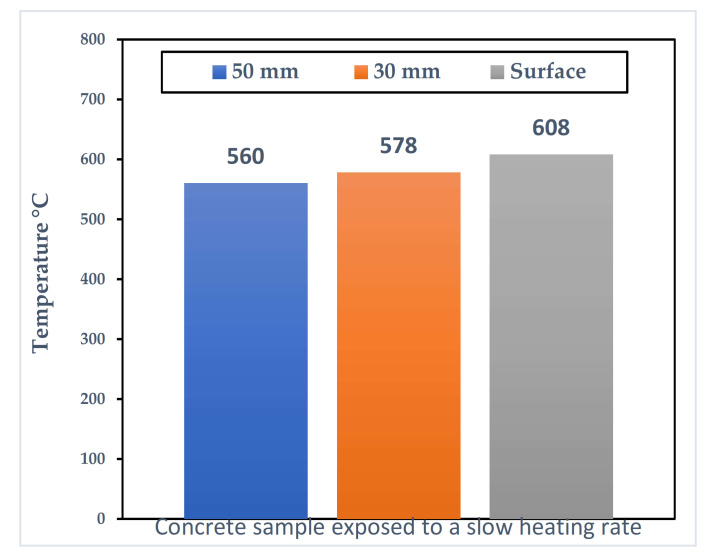
The temperature gradient after 12 h for a concrete sample exposed to a slow heating rate. Temperatures are recorded at the exposed surface, and 30 and 50 mm from the exposed surface.

**Figure 4 materials-15-01693-f004:**
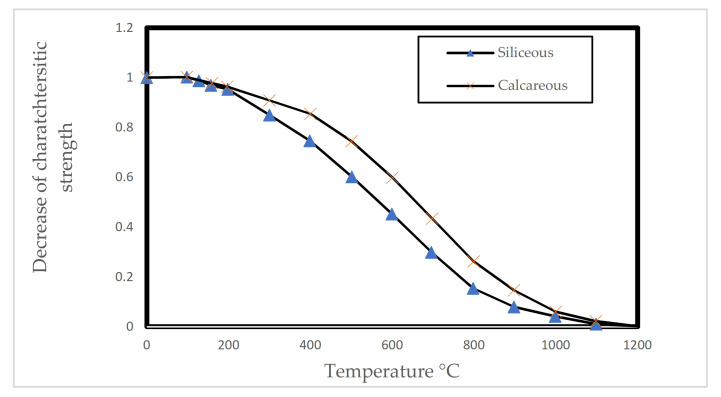
Decrease in concrete strength with rising temperature according to Eurocode 2.

**Figure 5 materials-15-01693-f005:**
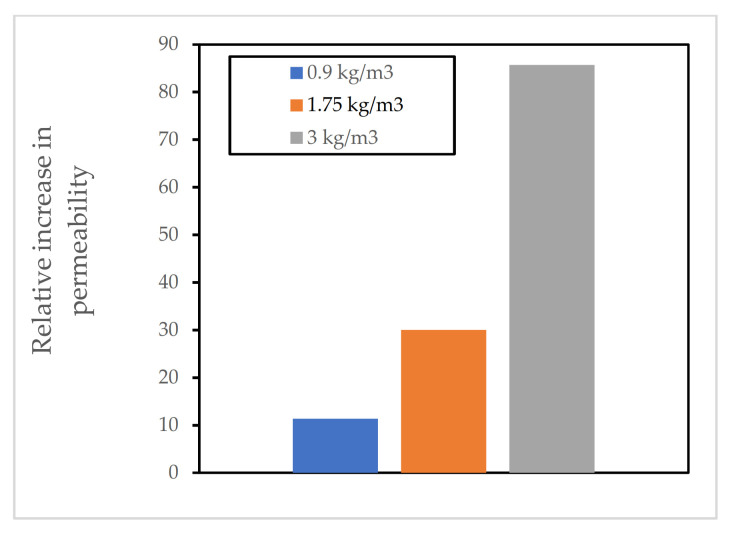
Increase in the relative permeability of concrete samples with PP fibres compared to samples without PP fibres at 200 °C.

**Figure 6 materials-15-01693-f006:**
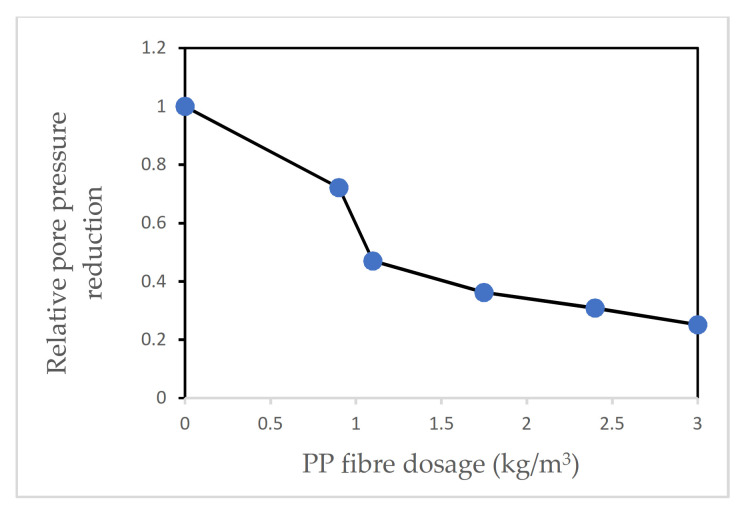
Relative reduction in the peak pore pressure with the increased dosage of PP fibres.

**Figure 7 materials-15-01693-f007:**
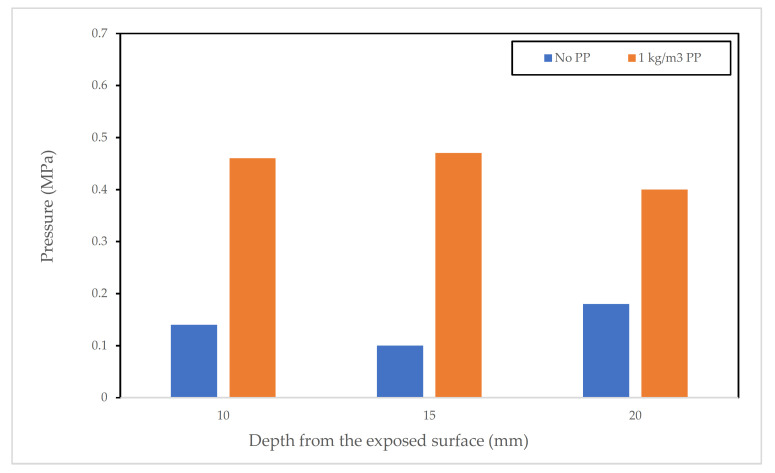
Peak pore pressure for SCC A and SCC B during exposure to HC fire. Slabs with PP fibre recorded higher peak pressure values than slabs without PP fibres.

**Figure 8 materials-15-01693-f008:**
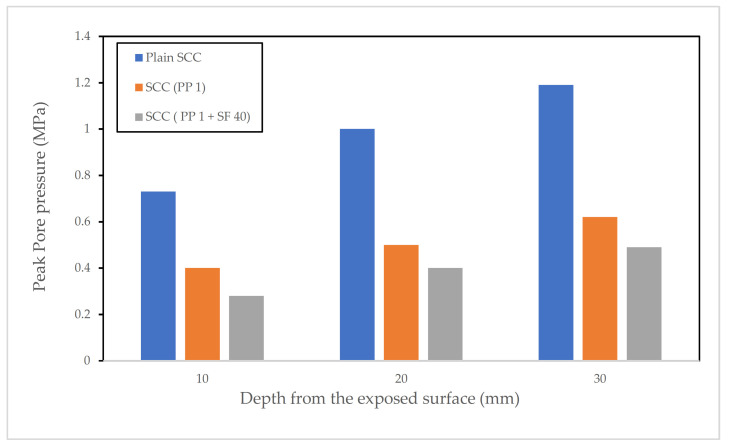
Peak pore pressure values at various depths from the exposed surface.

**Figure 9 materials-15-01693-f009:**
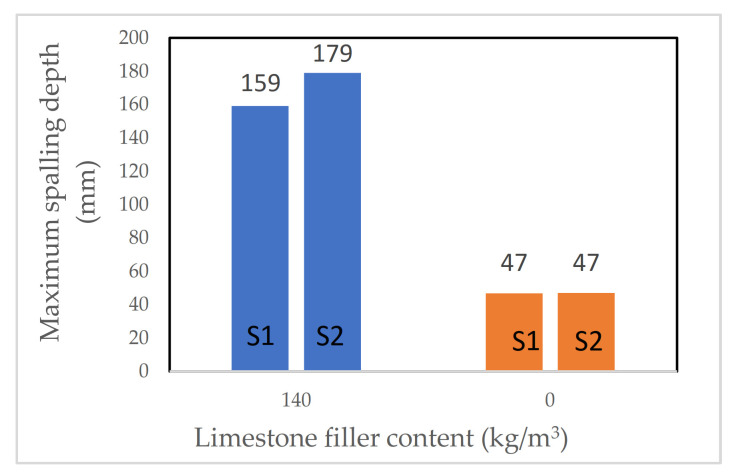
Maximum spalling depth as a function of the amount of limestone filler. For each concrete mix two samples (S1 and S2) were considered.

**Figure 10 materials-15-01693-f010:**
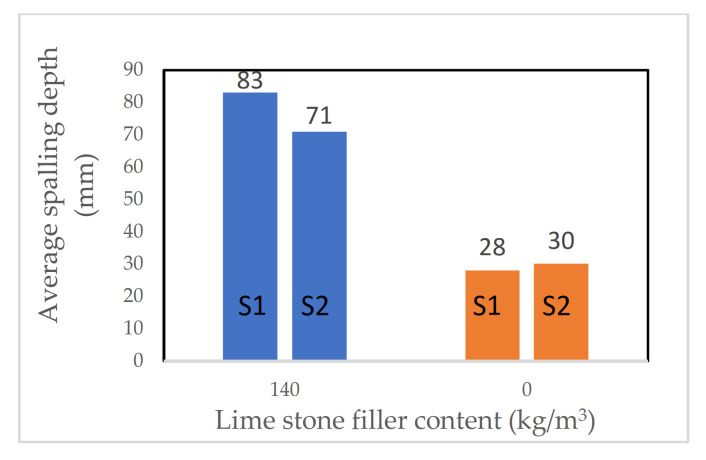
Average spalling depth as a function of the amount of limestone filler. For each mix two samples (S1 and S2) were considered.

**Figure 11 materials-15-01693-f011:**
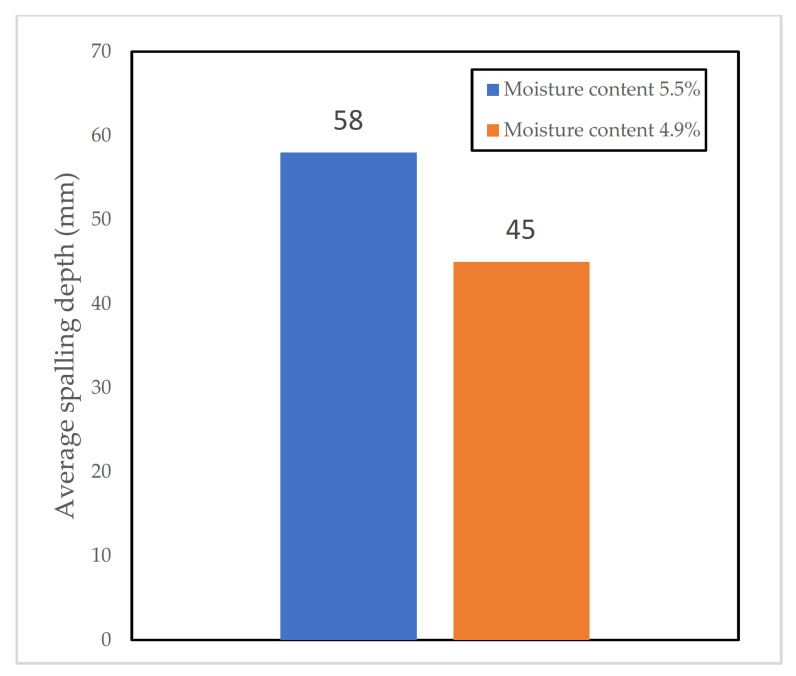
Average spalling depth as a function of the moisture content at the time of testing.

**Table 1 materials-15-01693-t001:** Heating rate, measurements type, and loading of the scientific research that studied the effects of type of concrete on its spalling of concrete as reported in the literature.

Paper	Heating Rate	Measurements	Loading
Khayat, 2014	NA	NA	NA
Boström, 2007	ISO 834	Visual inspection, temperature measurements, depth of spalling	Yes/No
Sideris, 2013	300 °C, 600 °C at 5 °C/min	Visual inspection, residual properties	No
Noumowé, 2006	ISO 834, Low heating rate 0.5 °C/min	Visual inspection, residual properties, temperature measurements	No
N. Anand, 2014	900 °C	Visual inspections, residual properties	No
B. Presson, 2004	ISO 834, HC	Visual inspections, mass loss, temperature measurements, Residual properties, water permeability, porosity	No
Bakhtiyari, 2011	ISO 834	TGA, XRD visual inspections, residual properties	No

**Table 2 materials-15-01693-t002:** Heating rate, measurements type, and loading of the scientific research that studied the effects of permeability on the spalling of concrete as reported in the literature.

Paper	Heating Rate	Measurements	Loading
Bošnjak, 2013	-	-	-
Jansson, 2010	ISO 834, RWS, HC	Visual inspections, temperature measurements, pore pressure	Yes
D. Dauti, 2018	500 °C	Visual inspection, temperature measurements, tomography imaging	No
Peng, 2018	800 °C at 10 °C/min	Visual inspection, XRD, porosity, GTA, SEM imaging	No
Kalifa, 2001	800 °C	Visual Inspection, pore pressure, gas permeability, temperature measurements, water permeability, SEM imaging	No
Zeiml, 2006	1 °C/min up to 600 °C	Permeability, porosity, SEM imaging	No
Miah, 2019	1 °C/min	Permeability	Yes
Bošnjak, 2013	0.5 °C/min	Permeability	Yes
D. Niknezhad, 2019	3 °C/min up to 500 °C	Visual inspection, permeability, residual properties	No

**Table 3 materials-15-01693-t003:** Heating rate, measurements type, and loading of the scientific research that studied the effects of PP on the spalling of concrete as reported in the literature.

Paper	Heating Rate	Measurements	Loading
Boström, 2008	ISO 834, HC, 10 °C/min	Visual inspection, depth of spalling	Yes
Jansson, 2010	ISO 834, RWS, HC	Visual inspections, temperature measurements, pore pressure	Yes
Jansson, 2013	HC, ISO 834	Visual inspection, temperature measurements, pore pressure	Yes
Terrasi, 2012	ISO 834	Visual inspection	Yes
Kalifa, 2001	800 °C	Visual inspection, pore pressure, gas permeability, temperature measurements, water permeability, SEM imaging	No
Maluk, 2015	ISO 834	Visual inspection, temperature measurements	Yes
Maluk, 2017	ISO 834	Visual inspection, temperature measurements, volume of spalling	Yes
Sultangaliyeva, 2017	ISO 834	Visual inspection, temperature measurements, volume of spalling	Yes
A. Al Qadi, 2014	5–10 °C/min	Visual inspection, residual properties	No
M. Uysal, 2012	800 °C	Visual inspection, residual properties	No
K. Sideris, 2013	5 °C/min up to 600 °C	Visual inspection, residual properties	No
Y. Ding, 2016	ISO 834	Visual inspection, residual properties, pore pressure, temperature measurements	No
Bangi, 2012	5 °C/min up to 600 °C	Visual inspection, pore pressure, temperature measurements	No
Noumowé, 2006	ISO 834, Low heating rate 0.5 °C/min	Visual inspection, residual properties, temperature measurements	No
P. Lura, 2014	ISO 834	Visual Inspection, Temperature measurements	Yes
Bošnjak, 2013	0.5 °C/min	Gas Permeability	Yes
Xargay, 2018	10 °C/min up to 600 °C	Visual inspection, residual properties	No
D. Zhang, 2018	ISO 834	Visual inspection, temperature measurements, gas permeability	No
Ye Li, 2019	ISO 834	Visual inspection, temperature measurements, gas permeability	No

**Table 4 materials-15-01693-t004:** Heating rate, measurements type, and loading of the scientific research that studied the effects of water/binder ratio on the spalling of concrete as reported in the literature.

Paper	Heating Rate	Measurements	Loading
Connolly, 1995	-	-	-
Morita, 2000	ISO 834	Visual inspections, temperature measurements, pore pressure, spalling depth, residual properties	Yes
Boström, 2008	ISO 834, HC, 10 °C/min	Visual inspection, depth of spalling	Yes
Boström, 2008	EN 1363-1	Visual inspection, depth of spalling, temperature measurements	Yes

**Table 5 materials-15-01693-t005:** Heating rate, measurement type, and loading of the scientific research that studied the effects of aggregate type on the spalling of concrete as reported in the literature.

Paper	Heating Rate	Measurements	Loading
Hager, 2018	ISO 834	Visual inspection, temperature measurement, depth of spalling	No
Khoury, 2011	-	-	-
A. Mohd Ali, 2018	HC	Visual Inspection, Depth of spalling	No
Xi Wu, 2018	10 °C/min up to 600 °C	Visual inspection, mass loss, ultrasonic pulse velocity, residual properties	No

**Table 6 materials-15-01693-t006:** Heating rate, measurement type, and loading of the scientific research that studied the effects of aggregate size on the spalling of concrete as reported in the literature.

Paper	Heating Rate	Measurements	Loading
Pan, 2012	5 °C/min up to 800 °C	TGA, visual inspection, mass loss	No
Y. Li, 2019	ISO 834	Visual inspection, gas permeability	No
A. Mohd Ali, 2018	HC	Visual inspection, depth of spalling	No

**Table 7 materials-15-01693-t007:** Heating rate, measurement type, and loading of the scientific research that studied the effects of concrete strength on its spalling as reported in the literature.

Paper	Heating Rate	Measurements	Loading
Sideris, 2013	5 °C/min up to 600 °C	Visual inspection, residual properties	No
Bakhtiyari, 2011	ISO 834	Visual inspection, XRD, mass loss, residual properties	No
Choe, 2015	ISO 834	Visual inspection, weight loss, residual properties, temperature measurements	No
Kalifa, 2001	800 °C	Visual inspection, pore pressure, gas permeability, temperature measurements, water permeability, SEM imaging	No
Mindeguia, 2013	ISO 834, 1 °C/min	Visual inspection, temperature measurement, pore pressure, gas permeability	No
Aslani, 2019	5 °C/min up to 900 °C	Visual inspection, weight loss, residual properties	No
Zheng, 2010	ISO 834	Visual inspection, depth of spalling, prestressing levels	Yes
Bošnjak, 2013	-	-	-

**Table 8 materials-15-01693-t008:** Heating rate, measurement type, and loading of the scientific research that studied the effects of externally induced stresses on the spalling of concrete as reported in the literature.

Paper	Heating Rate	Measurements	Loading
Terrasi, 2012	ISO 834	Visual inspection	Yes
Maluk, 2017	ISO 834	Visual inspection, temperature measurements, volume of spalling	Yes
Bonopera, 2022	-	Visual inspection, mechanical properties	Yes
Bošnjak, 2013	0.5 °C/min	Gas permeability	Yes
Miah, 2019	1 °C/min	Gas Permeability	Yes
Gan, 2019	-	Numerical analysis	-
Jansson, 2013	HC, ISO 834	Visual inspection, temperature measurements, pore pressure	Yes
Miah, 2017	-	Gas permeability	_
Zheng, 2010	ISO 834	Visual inspection, depth of spalling, prestressing levels	Yes
Rickards, 2020	ISO 834, HC	Visual inspection, temperature measurements	Yes

**Table 9 materials-15-01693-t009:** Heating rate, measurement type, and loading of the scientific research that studied the effects of heating rate on the spalling of concrete as reported in the literature.

Paper	Heating Rate	Measurements	Loading
Noumowé, 2006	ISO 834, Low heating rate 0.5 °C/min	Visual inspection, residual properties, temperature measurements	No
Mindeguia, 2013	ISO 834, 1 °C/min	Visual inspection, temperature measurement, pore pressure, gas permeability	No
Mindeguia, 2015	ISO 834, HC, slow heating rate, moderate heating rate	Visual inspection, temperature measurements, pore pressure, depth of spalling	No
Mindeguia, 2013	1 °C/min, 2 °C/min, 10 °C/min and 120 °C/min	Visual inspection, temperature measurements, pore pressure, residual properties	No
Phan, 2008	5 °C/min, 25 °C/min up to 600 °C	Visual inspection, pore pressure, gas permeability, temperature measurements	No
Choe, 2019	ISO 834, 1 °C/min	Visual inspection, weight loss, temperature measurements, pore pressure	No
Zhao, 2017	ISO 834, 5 °C/min	Numerical model analysis	No

**Table 10 materials-15-01693-t010:** Heating rate, measurement type, and loading of the scientific research that studied the effects of moisture content/sample age on the spalling of concrete as reported in the literature.

Paper	Heating Rate	Measurements	Loading
Connolly, 1995	-	-	-
Jansson, 2013	HC, ISO 834	Visual inspection, temperature measurements, pore pressure	Yes
Jansson, 2013	-	-	-
Mindeguia, 2015	ISO 834, HC, slow heating rate, moderate heating rate	Visual inspection, temperature measurements, pore pressure, depth of spalling	No
Maier, 2020	HC	Visual inspection, temperature measurements, gas permeability, depth of spalling	No
Choe, 2019	ISO 834, 1 °C/min	Visual inspection, weight loss, temperature measurements, pore pressure	No
Peng, 2018	800 °C at 10 °C/min	Visual inspection, XRD, porosity, GTA, SEM imaging	No

**Table 11 materials-15-01693-t011:** Heating rate, measurement type, and loading of the scientific research that studied the effects of silica fume/binder ratio on the spalling of concrete as reported in the literature.

Paper	Heating Rate	Measurements	Loading
Ahmad, 2019	3 °C/min	Visual inspection, residual properties	No
Bakhtiyari, 2011	ISO 834	Visual inspection, residual properties	No
Behnood, 2009	3 °C/min up to 600 °C	Visual inspection, residual properties	No
Ju, 2017	5 °C/min	Visual inspection, residual properties	No

**Table 13 materials-15-01693-t013:** Heating rate, measurement type, and loading of the scientific research that studied the effects of sample size on the spalling of concrete as reported in the literature.

Paper	Heating Rate	Measurements	Loading
Min Li, 2004	GB/T 9978-1999 (Similar to ISO 834)	Visual inspection, residual properties	No
Jansson, 2013	HC, ISO 834	Visual inspection, temperature measurements, pore pressure	Yes

**Table 14 materials-15-01693-t014:** Heating rate, measurement type, and loading of the scientific research that studied the effects of curing process on the spalling of concrete as reported in the literature.

Paper	Heating Rate	Measurements	Loading
Peng, 2018	800 °C at 10 °C/min	Visual inspection, XRD, porosity, GTA, SEM imaging	No
Jansson, 2013	HC, ISO 834	Visual inspection, temperature measurements, pore pressure	Yes
Turkmen, 2007	NA	Porosity measurements	Yes
Oliviera, 2015	NA	Visual inspections, spalling depth	No
Singh, 2013	NA	Mechanical properties	NA

**Table 15 materials-15-01693-t015:** Heating rate, measurement type, and loading of the scientific research that studied the effects of other types of fibres/additives on the spalling of concrete as reported in the literature.

Paper	Heating Rate	Measurements	Loading
Abdulhaleem, 2018	5 °C/min	Visual inspection, XRD, porosity, GTA, SEM imaging	No
Jansson, 2013	HC, ISO 834	Visual inspection, temperature measurements, pore pressure	Yes
Turkmen, 2007	NA	Porosity measurements	Yes
Oliviera, 2015	NA	Visual inspections, spalling depth	No
Han, 2011	ISO 834	Visual inspection, residual properties, weight loss	No

**Table 16 materials-15-01693-t016:** Heating rate, measurement type, and loading of the scientific research that studied the effects of air entrainment on the spalling of concrete as reported in the literature.

Paper	Heating Rate	Measurements	Loading
Khaliq, 2017	10 °C/min	Visual inspection, residual properties, mass loss	No
Drzymala, 2017	600 °C	Visual inspection, temperature measurements, residual properties	No
Holan, 2019	10 °C/min up to 800 °C	Visual inspection, residual properties	No
Oliviera, 2015	NA	Visual inspections, spalling depth	No
P. Lura, 2014	ISO 834	Visual inspection, temperature measurements	Yes

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
