# Peer review of "Heat-Induced Spalling of Concrete: A Review of the Influencing Factors and Their Importance to the Phenomenon"

_materials, 2022, doi:10.3390/ma15051693_

Round 1

Reviewer 1 Report

The manuscript “materials-1596751” presents an attractive review of the spalling mechanism of concrete exposed to high temperatures. The paper is recommended for publication as it is well written and deals with an interesting topic. However, some comments to improve the quality of the paper are below:

  1. In the 1st line of the abstract and on line 35, Spalling in concrete is a complicated problem, please revise it, different researchers can have different opinions on it, however, spalling is explainable with the help of science. In a scientific paper, we should not write it’s a complicated problem.
  2. In the abstract, the authors said -This paper, reports previously unpublished results on spalling of self-prestressed SCC- Authors have reviewed some of the published papers and did nothing experimental, how can they write that this paper reports previously unpublished results on spalling? Based on the review, you can suggest, recommend and advise something the readers of the journal. Please consider revising it.
  3. Authors are requested to improve section 5.4 -Type of aggregate, the influence of lightweight porous aggregate on spalling mechanism can briefly be explained here. Also, the effect of slate and expanded shale on spalling damage of concrete can be mentioned here.
  4. Finally, the effect of the addition of different polymeric resins, such as epoxy and acrylic to concrete on the surface spalling can be added as a separate section to improve the quality of this review paper.

Author Response

Dear Reviewer,
Enclosed please see the revised version of the manuscript and the response to your valuable remarks.

Reviewer 2 Report

The reviewer appreciates the “paper review” done by the authors. In his/her opinion, the goal of the work must be better explained within Abstract, Introduction and Conclusions. Moreover, the publication in the “Materials, MDPI” is not recommended unless the following suggestions are taken into account within the article:

1) In Section 5.7. “Externally induced stresses”, some references should be cited.

“Terrasi et al. [84] observed that spalling happened in the region where there was minimum bending moment (i.e., near the supports where compressive stress from prestressing would be greatest). This is reported by the authors to showcase the adverse effects of compressive stress on spalling.”
This behavior was also underlined in the literature for simply supported Prestressed Concrete (PC) members. Please, refer to this issue and cite the following reference:
-  Influence of prestressing on the behavior of uncracked concrete beams with a parabolic bonded tendon. Structural Engineering and Mechanics, 2021, 77(1), pp. 1–17.

“Miah et al. [21] investigated the effects of compressive loading on the permeability of ordinary concrete samples at elevated temperatures. Permeability studies were performed at room temperature on samples subjected to various preloading conditions and preheating to 80, 120, 250, 400, 600, and 800 C. The authors reported that the presence of load could lead to an increase or decrease of concrete permeability depending on the direction of the applied load; when the load was in the direction of gas flow (i.e., parallel to the cracks formed at the interface of aggregate and cement paste), permeability increases, but when the load was applied perpendicular to the direction of the gas flow (and the cracks), then permeability of the sample decreases.”
The permeability of concrete also increases in presence of shrinkage and/or creep due to the occurrence of cracking as, e.g., in PC members. Please, refer to this issue and cite the following references:
-  The effect of prestressing force on natural frequencies of concrete beams - A numerical validation of existing experiments by modelling shrinkage crack closure. J. Sound Vib. 2019 455 pp. 20–31.

2) In Section 5.13. “Curing”, some references should be cited.

Curing increases elastic modulus and compressive strength of concrete. Consequently, porosity of concrete is influenced with time. This aspect was treated in the literature for PC members. Please, refer to this issue and cite the following references:
-  Effect of a time dependent concrete modulus of elasticity on prestress losses in bridge girders. Int. J. Concr. Struct. Mater. 2013 7 (3) pp. 183–91.

3) There are several reasons why spalling occurs in concrete elements. However, the most common cause of spalling is the corrosion of embedded steel reinforcement bars or steel sections in prestressed and reinforced concrete. Please, insert a specific section regarding this issue with the corresponding most important references.

4) The paper could also be improved following two directions to give a clearer message to the reader that could be rather confused by the extension of writing and the number of references. In short, the reviewer finds this article rather lacks a precise line of discourse and therefore the reader reads a list of existing research without effectively capturing the message of the authors. (1) Introduce a table for each section where the literature about “spalling of concrete” is discussed to schematically characterize each reference (e.g. authors, year, lab or numerical experiments, topic, findings). (2) Introduce original figures with schemes to explain the driving ideas traced by the literature review. This also could help the reader to have a more straightforward idea of the material.

5) The paper must have a final section, i.e., “7. Conclusions”.

6) I suggest to the authors to edit all the text of the paper with the help of a native English speaker. Grammar, punctuation, spelling, verb usage, sentence structure, conciseness, readability and writing style could be improved.

Author Response

(The authors gave the same response as above.)

Round 2

Reviewer 2 Report

The authors have adequately addressed my comments.